# Forest-floor respiration, N$_2$O, and CH$_4$ fluxes in a subalpine spruce forest: Drivers and annual budgets

Luana Krebs[1], Susanne Burri[1], Iris Feigenwinter[1], Mana Gharun[2], Philip Meier[1], Nina Buchmann[1]

[1]Department of Environmental Systems Science, Institute of Agricultural Sciences, ETH Zurich, Switzerland
[2]Faculty of Geosciences, Institute of Landscape Ecology, University of Munster, Germany

*Correspondence to*: Luana Krebs (luana.krebs@usys.ethz.ch)

**Abstract**. Forest ecosystems play an important role in the global carbon (C) budget by sequestering a large fraction of anthropogenic carbon dioxide (CO$_2$) emissions and by acting as important methane (CH$_4$) sinks. The forest-floor greenhouse gas (GHG; CO$_2$, CH$_4$ and nitrous oxide N$_2$O) flux, i.e., from soil and understory vegetation, is one of the major components

to consider when determining the C or GHG budget of forests. Although winter fluxes are essential to determine the annual C budget, only very few studies have examined long-term, year-round forest-floor GHG fluxes. Thus, we aimed to i) quantify seasonal and annual variations of forest-floor GHG fluxes; ii) evaluate their drivers, including the effects of snow cover, timing, and amount of snowmelt, and iii) calculate annual budgets of forest-floor GHG fluxes for a subalpine spruce forest in Switzerland. We measured GHG fluxes year-round during four years with four automatic large chambers at the ICOS Class 1

Ecosystem station Davos (CH-Dav). We applied random forest models to investigate environmental drivers and to gap-fill the flux time series. The forest floor emitted 2336 g CO$_2$ m$^{-2}$ yr$^{-1}$ (average over four years). Annual and seasonal forest-floor respiration responded most strongly to soil temperature and snow depth. No response of forest-floor respiration to leaf area index or photosynthetic photon flux density was observed, suggesting a strong direct control of soil environmental factors and a weak or even lacking indirect control of canopy biology. Furthermore, the forest-floor was a consistent CH$_4$ sink (-0.71 g

CH$_4$ m$^{-2}$ yr$^{-1}$), with annual fluxes driven mainly by snow depth. Winter CO$_2$ fluxes were less important for the CO$_2$ budget (6.0–7.3 %), while winter CH$_4$ fluxes contributed substantially to the annual CH$_4$ budget (14.4–18.4 %). N$_2$O fluxes were very low (0.007 g N$_2$O m$^{-2}$ yr$^{-1}$), negligible for the forest-floor GHG budget at our site. In 2022, the warmest year on record with below-average precipitation at the Davos site, we observed a substantial increase in forest-floor respiration compared to other years. The mean forest-floor GHG budget indicated emissions of 2319±200 g CO$_2$-eq m$^{-2}$ yr$^{-1}$ (mean ± standard deviation over

all years), with respiration fluxes dominating and CH$_4$ offsetting a very small proportion (0.8 %) of the CO$_2$ emissions. Due to the relevance of snow cover, we recommend year-round measurements of GHG fluxes with high temporal resolution. In a future with increasing temperatures and less snow cover due to climate change, we expect increased forest-floor respiration at this subalpine site modulating the carbon sink of the forest ecosystem.

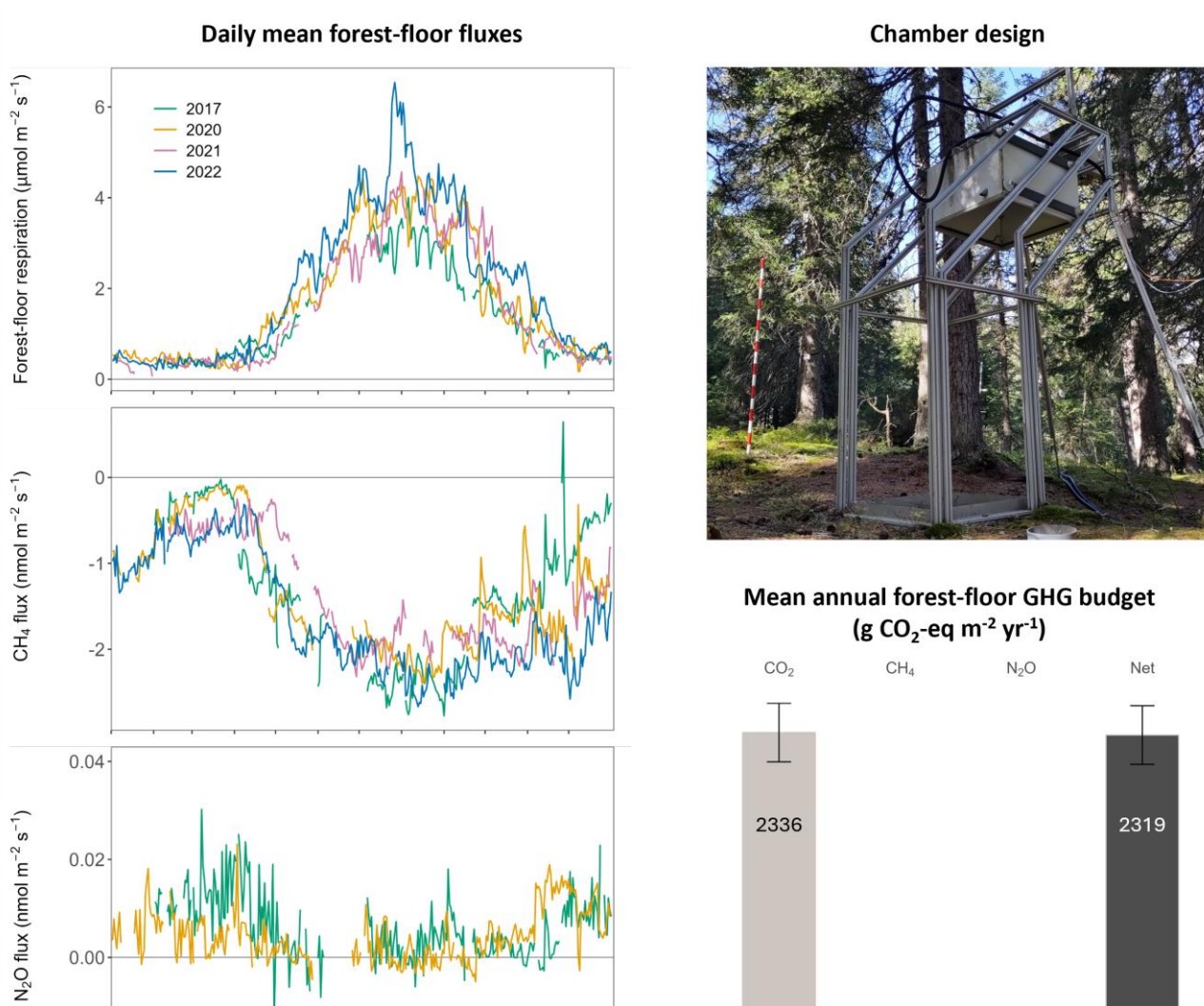

**Daily mean forest-floor fluxes**

**Chamber design**

**Mean annual forest-floor GHG budget (g CO$_2$-eq m$^{-2}$ yr$^{-1}$)**

## 1 Introduction

Carbon dioxide (CO$_2$), methane (CH$_4$), and nitrous oxide (N$_2$O) are the three main greenhouse gases (GHGs) driving global warming. Forest ecosystems play an important role in the global carbon (C) cycle by sequestering a large fraction of anthropogenic CO$_2$ emissions and by acting as an important CH$_4$ sink (Borken et al., 2006; Ni and Groffman, 2018; Friedlingstein et al., 2023). The GHG flux of the forest-floor, i.e., soil and understory vegetation, is one of the major components to consider when determining the C budget of forests, since soil respiration is the second largest terrestrial C flux

and accounts for approximately 70 % of $CO_2$ losses in temperate forests (IPCC, 2021; Yuste et al., 2005). However, how forest-floor GHG fluxes will respond to climate change is still largely unknown.

Global warming particularly affects high latitude and high altitude forests (IPCC, 2021), altering snowfall, length and timing of snow cover as well as melting and soil freeze-thaw cycles (CH2018, 2018; Klein et al., 2016). Nevertheless, there have been very few studies that examined continuous, year-round and long-term forest-floor GHG fluxes in high latitude or high altitude forests (Barba et al., 2019; Luo et al., 2011). Unfortunately, measurements during periods with snow cover are challenging and thus often lacking due to logistical reasons, leading to winter fluxes missing even from multi-year studies (e.g., Richardson et al., 2019).

Forest-floor $CO_2$ fluxes include photosynthetic $CO_2$ uptake by plants as well as autotrophic and heterotrophic respiratory losses from plants and soils, respectively (Hanson et al., 2000). All three processes and their contributions to the total soil $CO_2$ fluxes depend on biotic and abiotic factors as well as their interactions. For example, soil respiration is coupled to canopy photosynthesis and thus to incoming radiation, but also strongly controlled by soil conditions (i.e., soil temperature and moisture), substrate availability, and the microbial community (e.g., Högberg et al., 2001; Janssens et al., 2001; Scott-Denton et al., 2006). Furthermore, winter dynamics can impact soil respiration rates through changes in snow cover, soil freezing and thawing cycles (Reinmann and Templer, 2018; Schindlbacher et al., 2007). Especially, freeze-thaw events have recently been the focus of research because they cause abrupt changes in biophysical soil conditions which can alter autotrophic and heterotrophic soil respiration rates (Song et al., 2017). How soil respiration responds to climate change is, however, not fully clear. With increasing soil temperatures as a consequence of increasing air temperatures (Lembrechts et al., 2022), global observations and models show a globally rising trend of soil respiration over recent decades and a continuation of the increase with progressing climate change (Bond-Lamberty et al., 2018; Nissan et al., 2023). At the same time, there is evidence for a thermal optimum of ecosystem respiration over a range of different biomes, indicating a non-monotonic relationship between soil temperature and respiration (Chen et al., 2023).

Forest soils have been shown to act as an atmospheric $CH_4$ sink (Dutaur and Verchot, 2007). The uptake of $CH_4$ in well-aerated soils is related to the presence of methane-oxidizing bacteria (Saunois et al., 2020). This process is highly dependent on environmental factors, including soil temperature ($T_{soil}$), soil texture (transport of $CH_4$ into the soil), soil moisture (transport of $CH_4$ into the soil and limitation of bacterial activity), and soil nitrogen (N) content (Borken et al., 2006; Luo et al., 2013; Ni and Groffman, 2018). Furthermore, biotic factors such as plant cover can affect $CH_4$ uptake of the forest floor through the presence of *Sphagnum* moss species which are inhabited by methane-oxidizing bacteria (Basiliko et al., 2004). Generally, in temperate forests, $CH_4$ uptake increases in warmer and drier soils (Borken et al., 2006; Ni and Groffman, 2018). Winter dynamics further impact $CH_4$ fluxes, with frozen soil and snow cover affecting microbial activity and gas transport (Blankinship et al., 2018; Borken et al., 2006). Understanding the drivers of forest-floor $CH_4$ fluxes, including the complex interplay between biotic and abiotic factors, is vital for accurately modeling and predicting the role of forest ecosystems in the global $CH_4$ cycle.

Moreover, the forest-floor can act as a source or sink of $N_2O$ (Chapuis-Lardy et al., 2007; Goldberg et al., 2010). Soil temperature, soil moisture, and N availability significantly influence $N_2O$ fluxes through regulating microbial processes which are mainly responsible for $N_2O$ production in soils, i.e., nitrification and denitrification (Schaufler et al., 2010). High $N_2O$ emission rates in temperate forests have been found under warm and moist conditions (Luo et al., 2013). Furthermore, high $N_2O$ emissions occur during freezing-thawing cycles and rewetting events, when abrupt changes in temperature and moisture conditions promote microbial activity and thus the release of $N_2O$ (Goldberg et al., 2010; Papen and Butterbach-Bahl, 1999; Liu et al., 2018; Butterbach-Bahl et al., 2013). Understanding the dynamics of these processes and drivers, particularly during freezing-thawing cycles, is crucial for estimating $N_2O$ emissions from forests.

In this study, we investigated combined measurements of forest-floor respiration, $CH_4$ and $N_2O$ fluxes in a subalpine Norway spruce forest (Davos, CH-Dav, ICOS Class 1 Ecosystem station), in response to biotic and environmental drivers, based on four years of year-round measurements (2017, 2020-2022). Our objectives were to i) quantify seasonal and annual variations in climate variables and forest-floor respiration, $CH_4$ and $N_2O$ fluxes; ii) evaluate the drivers of forest-floor GHG fluxes, including effects of snow cover, timing and amount of snowmelt; and iii) calculate the annual budgets of forest floor GHG fluxes. We hypothesized that the forest floor is a source of $CO_2$ throughout the years, with large seasonal variability due to the temperature sensitivity of respiratory processes, but very low $N_2O$ emissions due to the overall low N supply at the site. In contrast, we expected that the forest floor is a net sink of $CH_4$, with soil temperature and snow dynamics being important drivers due to their impact on microbial activity and diffusion rates between soil and atmosphere. Thus, we expected the highest respiratory fluxes and $CH_4$ uptake in 2022, an exceptionally warm year at our site. Overall, we anticipated the forest-floor GHG budget being mainly determined by respiration fluxes, with $CH_4$ uptake only slightly offsetting the respiratory $CO_2$ losses and $N_2O$ emissions being negligible.

## 2 Methods

### 2.1 Study site

The study site is a subalpine evergreen coniferous forest, located in the eastern Swiss Alps at an altitude of 1640 m a.s.l. (Davos Seehornwald; CH-Dav; 46°48'55.2" N, 9°51'21.3" E). The total annual precipitation is 876 mm, and the mean annual temperature is 4.3 °C (1997–2022). The site is certified as ICOS (Integrated Carbon Observation System) Class 1 Ecosystem station for eddy-covariance flux measurements since 2019. The dominant species is Norway spruce (*Picea abies* (L.) Karst), with an average tree height of 18 m (max. 35 m), and a mean tree age of approx. 100 years (with some trees reaching over 300 years). Understory vegetation covers about 30 % of the surface and is mainly composed of blueberry (*Vaccinium myrtillus* and *Vaccinium gaulterioides*) and mosses (*Sphagnum* sp. Ehrh. and *Hylocomium splendens*). CH-Dav is a sustainably managed forest according to the Swiss National Forest Protection Law (1876; Tschopp, 2012). The soil types are chromic cambisols (L, F, H layers with 1 cm, 2 cm, and 1.5 cm thickness, respectively; A, B(h), B(fe), B, and (B)Cv horizons with 0–4 cm, 4–12 cm, 12–45 cm, 45–70 cm, and > 70 cm depth, respectively) and rustic podzols (L, F, H layers with 1 cm, 3 cm, and 7 cm thickness,

respectively; Ah, (A)E, Bfe, BCv, and (B)Cv horizons with 0–3 cm, 3–10 cm, 10–40 cm, 40–80 cm, and > 80 cm soil depth, respectively; FAO classification; Jörg, 2008). Soil texture ranges from sand to sandy loam (Jörg, 2008). Soil bulk density at 5 cm depth of mineral soil is between 0.27–0.35 g cm$^{-3}$ (Saby et al., 2023). Soil C and N stocks (0 to 60 cm depth) are on average 142.3 and 5.1 t ha$^{-1}$, respectively (Jörg, 2008).

## 2.2 Chamber flux measurements

### 2.2.1 Chamber set-up and tests

Forest-floor respiration, $CH_4$ and $N_2O$ fluxes were measured during the years 2017 and 2020–2022 using a fully automated system with four chambers (FF1 to FF4), distributed within an area of 3600 m$^2$ in the forest to represent the eddy-covariance footprint. Concentrations of $CO_2$, $CH_4$ and $N_2O$ were measured with a Dual Laser Trace Gas Analyzer (TILDAS, Aerodyne Research, Billerica, USA) since 2017. To ensure high measurement quality, laser temperatures and tuning rates were adjusted on a regular basis. After the measurement campaign in 2017, the TILDAS was sent to Aerodyne for maintenance and repair (new $N_2O$ laser source); measurements of all three GHG were resumed in fall 2019. In January 2021, the $N_2O$ laser broke, thus $N_2O$ measurements stopped. Since November 2019, $CO_2$ concentrations in the chambers were also measured with an infrared gas analyzer (LI-840, LI-COR Biosciences, Lincoln NE, USA). For the year 2020, $CO_2$ chamber measurements from both TILDAS and LI-840 were available and used for further analyses (see below). For 2021 and 2022, only the IRGA $CO_2$ measurements were used.

Chambers were designed according to Brümmer et al. (2017), following the ICOS RI protocol for chamber measurements (Pavelka et al., 2018). The large opaque PVC chambers (white surfaces to increase albedo) rested on aluminum frames, and were inserted 10 cm into the soil, sealed with EPDM (ethylene propylene diene monomer) gaskets. Their large size (75 cm x 75 cm x 50 cm height, approx. 281 dm$^3$) allowed to reduce edge effects as much as possible. Chambers were equipped with a pressure vent, as well as air temperature and pressure sensors (BME280, Bosch Sensortec GmbH, Reutlingen, Germany). During the winter periods with snowfall, extension frames (2 x 50 cm height) allowed to increase chamber height. A 17 Watt geared electric motor (80807021, Crouzet, Valence, France) was used to move the entire PVC chamber vertically and horizontally by about 190 cm and 70 cm, respectively (Fig. A.1). One webcam per chamber allowed remote observation of the operation and estimates of snow cover and depth (see below). Since the vegetation inside the chamber frames was not cut, the chamber set-up measured forest-floor GHG fluxes (and not only soil fluxes). Due to their opaque material, no understory photosynthesis was measured with the chambers. Soil and vegetation cover inside the chambers (differentiated into three plant functional types: moss, grass, blueberry) were assessed visually in June 2022, when also leaf area index (LAI) of the spruce forest was measured using digital photography above the chamber locations (Fuentes et al., 2008). One chamber cycle lasted 10 minutes, with 3 minutes for the actual measurement period (when chamber resided on the frame, i.e. was fully closed), 3.5 minutes for closing, and 3.5 minutes for opening the chambers (slow upward and sideward movement was controlled by an Arduino Ethernet). Thus, chambers were fully closed for 48 minutes per day (i.e., 3.3 % of the day), during which rainfall was

fully excluded. If we added the time when the chambers were hovering directly over the frame (about 4 minutes per cycle), the chambers would be closed for a maximum of 7 minutes per chamber cycle (i.e., 7.8 % of the day). But this is a rather

conservative estimate of rainfall exclusion, since rain does not always fall vertically, and throughfall is typically much less than bulk precipitation due to canopy interception. Together with further tests on potential chamber effects, i.e., SWC inside vs. outside for two chambers and four years; $T_{soil}$ inside vs. outside for four chambers and three years (see Appendix, Figs. A.2-4), we concluded that our chamber design and closure duration avoided potential effects on environmental conditions as much as possible.

During the 10 min cycles, concentrations were measured continuously once per second. The air from the chamber was fed to the gas analyzers in 6 mm OD tubing (Synflex 1300, Eaton, Dublin, Ireland) and pumped back to the chamber, forming a closed system. Tube lengths between chamber and instrument ranged between 49–85 m, and the flow rate ranged between 0.9–1.0 slpm. We determined the time lags for the arrival of gas in the instrument based on the change in chamber status (fully open, fully closed) and max. $CO_2$ concentrations measured. Switching of the air stream between different chambers and gas

analyzers was accomplished using rotary selector valves (Valco Selectors, VICI AG International, Schenkon, Switzerland). Chamber cycles (lasting app. 1 h for four chambers) were repeated every three hours for each gas analyzer individually, leading to a total of 16 cycles per chamber and day (eight per gas analyzer). Leakage tests of all four chambers were performed in 2019. Variations caused by possible leakage were below 3% of the measured flux, as required by the ICOS RI protocol (Pavelka et al., 2018).

## 2.2.2 Data processing and quality assessment

The concentration increase in the chamber headspace over time was used to determine the respective flux F using Eq. (1):

$$F = \frac{\frac{\partial C}{\partial t} \frac{V}{A} \frac{m}{V_m} \frac{p}{p_0} \frac{T_0}{T}}{m} \tag{1}$$

where $\frac{\partial C}{\partial t}$ is the concentration change over time (mol mol$^{-1}$ s$^{-1}$), $V$ the actual chamber volume (m$^3$), $A$ the forest-floor area within the chamber frame (m$^2$), $m$ the molecular mass (dimensionless), $V_m$ the molar volume (m$^3$ mol$^{-1}$) of the respective gas,

$p$ the mean chamber pressure (Pa), $p_0$ the standard pressure (1013.25 Pa), $T_0$ the standard temperature (273.15 K), and $T$ the mean chamber temperature (K). We accounted for the varying chamber volume due to snow depth and additional extension frames installed during winter. Thus, the actual chamber volume was calculated using Eq. (2):

$$V = A \times (h_{chamber} + h_{frame} - h_{snow}) \tag{2}$$

where $h_{chamber}$ is the height of the chamber, $h_{frame}$ the height of the extension frame(s), and $h_{snow}$ the snow depth. We fitted

a linear regression to the change in concentration of the respective gas over time ($\frac{\partial C}{\partial t}$) during the closed period of the chamber (180 s), excluding the first 20 s after closing. The R$^2$ and root mean square error (RMSE) of the fit was later used for the quality assessment and filtering of the calculated fluxes (see below). A positive flux means release from the forest floor to the atmosphere, and a negative flux indicates uptake by the forest floor.

The quality of the calculated fluxes was ensured by removing negative $CO_2$ fluxes (Step 1), removing outliers (Step 2, despiking), and applying a filter based on $R^2$ for $CO_2$ and $CH_4$ and based on RMSE for $N_2O$ (Step 3). These three steps were applied to each GHG separately. In more detail: (1) We excluded negative $CO_2$ fluxes (about 2 % of all fluxes). (2) We then despiked (i.e., removed outliers) the flux data set with a running mean algorithm using a width of 30 days. Step 2 removed 0.2 %, 0.7 % and 1.2 % of $CO_2$, $CH_4$ and $N_2O$ fluxes, respectively. (3) For $CO_2$ and $CH_4$ fluxes, we analyzed data separately for each gas, each chamber, as well as growing period (May to November) vs. dormant period (December to April), and based the quality assessment on $R^2$ values. We removed fluxes with a $R^2$ value below the 10th percentile of all $R^2$ values in the respective period (except if $R^2 > 0.9$), to avoid setting a fixed threshold for an acceptable $R^2$. The 10th percentile of $R^2$ values ranged from 0.21 to 0.99, being lower during the dormant compared to the growing period (Tab. A.1). Step 3 excluded 6 % and 9 % of $CO_2$ and $CH_4$ fluxes, respectively. For $N_2O$ fluxes, we separated data of the two years available (2017, 2020) to account for the replacement of the $N_2O$ laser source in 2019 and based the quality assessment on the RMSE (due to the low magnitude of the $N_2O$ fluxes). $N_2O$ fluxes with an RMSE below the 10th percentile of all RMSE, which were 0.13 and 0.03 for 2017 and 2020, respectively, were removed. Step 3 excluded 25 % of all $N_2O$ fluxes. Furthermore, for $N_2O$, we estimated a minimum reliable flux with the specifications of the TILDAS instrument (precision of 0.03 ppb) and the closure time, i.e., any change of $N_2O$ concentrations in the chamber headspace during the measurement period had to be $> 0.06$ ppb (McManus et al., 2006) or $>$ 29.1 nmol $N_2O$ m$^{-2}$ h$^{-1}$.

Overall, the initial time series consisted of 38'103 $CO_2$ (in 2020 from two gas analyzers), 27'503 $CH_4$ and 13'291 $N_2O$ flux measurements over the four years. After the quality checks described above, 34'938 $CO_2$ (92 %), 25'083 $CH_4$ (91 %), and 9'823 $N_2O$ (74 %) flux measurements remained, which resulted in 4446, 3972, and 1755 daily means, respectively.

### 2.2.3 Static chamber measurements

In order to check for the validity of our $N_2O$ flux measurements using the automatic chambers, we performed $N_2O$ measurements using static chambers (dimensions of d = 30 cm and h = 30 cm; Hutchinson and Mosier, 1981). We used eight static chambers, i.e., four chambers next to the automatic chambers, and four chambers placed randomly within the research area. Soil collars were installed two weeks prior to the first measurement campaign. Four rounds of measurements were done on two days in October 2023 (n=32), when soil temperature was between 5.5-12 °C, well above the long-term mean, and when WFPS at 5 cm depth were on average 13.1 %, favoring microbial activities. Three collars were irrigated between the first and second measurement round on the two days to simulate a heavy rainfall event, favoring denitrification. We left the chambers closed for 1 h and sampled air in the headspace every 20 min. Sampling and flux calculations were done as described in Barthel et al. (2022). All gas samples were analyzed at ETH Zurich for $N_2O$ mole fraction using gas chromatography (456-GC, Scion Instruments, UK).

**2.3 Environmental data**

Each of the chambers had measurements of soil water content (SWC; EC-5, Decagon Devices Inc.) and $T_{soil}$ (107, Campbell Scientific Ltd.) at 5 cm soil depth in close vicinity (< 2 m away from the chamber). To account for potential drivers of canopy photosynthesis modulating forest-floor fluxes, photosynthetic photon flux density (PPFD; PAR LITE, Kipp & Zonen), air temperature (TA; HygroClip HC2-S3, Rotronic AG), and precipitation (PREC; 1518H3, Lambrecht Meteo GmbH) data were used as well, measured at the tower above the tree canopy at 35 m height (precipitation measured at 25 m height).

We calculated water-filled pore space (WFPS) from the SWC measurements using Eq. (2):

$$WFPS = \frac{SWC}{1 - \frac{BD}{PD}} \times 100 \tag{2}$$

Bulk density (BD) was calculated using the data from a soil sampling campaign done in July 2018 according to ICOS RI standards (Arrouays et al., 2018; Saby et al., 2023). Soil data were used from soil profiles closest to the respective chambers (in total, data from six profiles were used). Particle density (PD) was assumed to be constant at 2.65 g cm$^{-3}$ (Danielson and

205 Sutherland, 2018). Mean daily snow cover and snow depth per chamber were derived from webcam images using a custom-made python image analysis tool, deriving snow depth from a scale installed in vicinity to each chamber within the image section.

**2.4 Statistical analyses**

**2.4.1 Driver analysis**

We used conditional random forests (RF) to model daily forest-floor respiration and CH$_4$ fluxes (based on all years and chambers) and investigated their environmental drivers. Due to the low N$_2$O fluxes, we excluded them from the RF analyses. We selected predictors which were known from the literature, i.e., daily averages of $T_{air}$, $T_{soil}$ at 5 cm depth, WFPS at 5 cm depth, and PPFD as well as their one- and four-day leads (meaning that we shifted the variables forward in time by one and four days). Furthermore, we added snow depth and changes in snow depth from one day to another (Δ snow depth) to the

predictor set. To account for factors which could explain differences in GHG fluxes among chambers, we included several chamber-specific characteristics (Tab. A.2), i.e., LAI, bare soil fraction in the chambers, and total C and N stocks in the topsoil (litter, organic layers, 0–20 cm depth of mineral soil). We applied the function *cforest* from the R-package "party" which can deal with highly correlated predictor variables (v1.3.10; Strobl et al., 2008, 2007). Prior to model development, predictors and target variables were centered and scaled using the "caret" *preProcess* function, which brings all variables and measurements

from different sensors and locations into the same range, improving performance of the RF models (v6.0.93; Kuhn, 2008). The hyperparameter fitting was done using the train function from the R-package "caret" (see Appendix for final model set-up) using 10-fold cross-validation. The assessment of driver importance in the RF model was done using the R package "permimp" which accounts for correlated variables within the predictor set (v.1.0.2; Strobl et al., 2007; Debeer and Strobl, 2020; Debeer

et al., 2021). The calculated values for driver importance were rescaled to values between 0 and 1 using a min-max
normalization.

We developed RF models separately for daily $CO_2$ and $CH_4$ fluxes (N = 4446 and 3972, respectively). The training of the RF
was done using only a fraction of the data set (70 %). The remaining 30 % of the data set was used as test dataset to evaluate
model performance. Centering and scaling were done separately for training and test datasets to avoid data leakage. The
performance of RF models was assessed using $R^2$ and RMSE. During model development, we tested several different predictor
sets. Furthermore, to optimize the models and to evaluate the robustness of model results, we evaluated the RF models trained
on data sets separated by year of measurement or by chamber, and compared their accuracy to the model that was built using
data from all years and all chambers. In total, 17 predictor variables entered the models (including the leads). RF models were
also trained on seasonal data (i.e., spring, summer, autumn, winter; defined according to the meteorological definition) to
investigate differences in drivers among seasons. For seasonal RFs, we used the same predictor sets as for the RFs developed
on the multi-year data set. We calculated partial dependence (PD) plots of conditional RFs using the "moreparty" package
(v0.3.1; Robette, 2023) which is based on the "pdp" package (v0.8.1; Goldstein et al., 2015; Greenwell, 2017) to assess
relationships between the four most important predictors and the predictions. The PD was calculated as the change in the
average predicted value, while the predictor of interest was varied over its marginal distribution.

**2.4.2 Flux gap-filling and budget calculations**

The gap-filling of $CO_2$ and $CH_4$ fluxes was done using the RF models described above. Missing values in the predictor variables
(gap length < 3 days) were linearly interpolated using the R package "chillR" (v0.72.8, Luedeling and Fernandez, 2022). The
gap-filled flux data were then used to calculate annual budgets of forest-floor C fluxes per chamber. Since we estimated the
annual forest-floor C budget for the study area, we reported the mean over the four chambers. To be able to compare $CO_2$, $CH_4$
and $N_2O$ budgets, we converted $CH_4$ and $N_2O$ fluxes into $CO_2$-equivalents ($CO_2$-eq) using the 100-year global warming
potential of 27 for methane, and 273 for $N_2O$ (IPCC, 2021).

In addition, we modeled the daily forest-floor respiration fluxes using a $Q_{10}$ model according to Eq. (3):

$$R_S = R_{ref} \times Q_{10}^{\frac{T_{soil} - 10}{10}}$$
(3)

where $R_{ref}$ is the modeled $R_S$ at $T_{soil}$ of 10°C, and $Q_{10}$ is the temperature sensitivity. We developed one model for the full dataset
(all years and all four chambers together). The annual respiration budgets calculated with the $Q_{10}$ modeled fluxes were then
compared to the annual respiration budgets from the RF gap-filling. All statistical analyses were performed using R Statistical
Software (v4.2.0, R Core Team, 2022).

# 3 Results

## 3.1 Seasonal and interannual variability of environmental conditions and GHG fluxes

The seasonal courses of $T_{air}$ and $T_{soil}$ were very pronounced during the four years of the study, with highest temperatures in July and August, and lowest temperatures in January (Fig. 1a). All years showed highly variable WFPS with large differences among chambers (i.e., up to 35 % difference; Fig. 1b), and highest WFPS values during the snowmelt period, i.e., March to May. While the snow-covered periods usually started in November and lasted until April or May (Fig. 1c), the snow depths were much higher during winter 2020/2021 (reaching snow depths of over 1 m) compared to the other winters. Overall, the year 2022 was the warmest year ever recorded at the Davos research site so far, with an annual mean $T_{air}$ of 5.6 °C (vs. the long-term mean of 4.3 °C; station data 1997–2022). Accordingly, annual mean $T_{soil}$ at 5 cm was highest in 2022 for all chambers (annual mean $T_{soil}$ over all chambers was 5.0 °C; Tab. A.2). At the same time, precipitation in 2022 was low (773 mm vs. long-term mean of 876 mm; station data 1997–2022), which led to comparably dry soil conditions (annual mean WFPS over all chambers was lowest in 2022 with 14.9 %).

The forest floor at the Davos Seehornwald site was a source of $CO_2$ during all four years, independent of the season (Fig. 1d). Typically, forest-floor respiration fluxes were very low in winter (mean $CO_2$ flux ± standard deviation (SD): 0.46±0.14 µmol $m^{-2}$ $s^{-1}$), increased in spring after the snowmelt, and reached their maximum values in June to September (mean $CO_2$ flux over all years of 3.50±0.84 µmol $m^{-2}$ $s^{-1}$). Lowest forest-floor respiration was measured in January 2021 (min. $CO_2$ flux of 0.06 µmol $m^{-2}$ $s^{-1}$), highest respiratory fluxes were observed in July 2022 (max. $CO_2$ flux of 6.54 µmol $m^{-2}$ $s^{-1}$).

Moreover, the forest floor was a consistent sink for $CH_4$, despite large short-term variations (days to weeks; Fig. 1e) and a few short peaks of $CH_4$ emissions in winter and spring. Seasonality of forest-floor $CH_4$ fluxes was very pronounced, with highest uptake in summer (mean $CH_4$ flux of -2.11±0.28 nmol $m^{-2}$ $s^{-1}$), and still high $CH_4$ uptake rates during autumn and early winter (October to December; most clearly seen in 2022). With increasing duration of winter (January to March; Fig. 1e), the $CH_4$ sink strength decreased, with lowest $CH_4$ uptake measured in March (mean $CH_4$ flux of -0.17±0.07 nmol $m^{-2}$ $s^{-1}$). However, after snowmelt, between April and end of May (depending on the year), $CH_4$ uptake rates increased sharply.

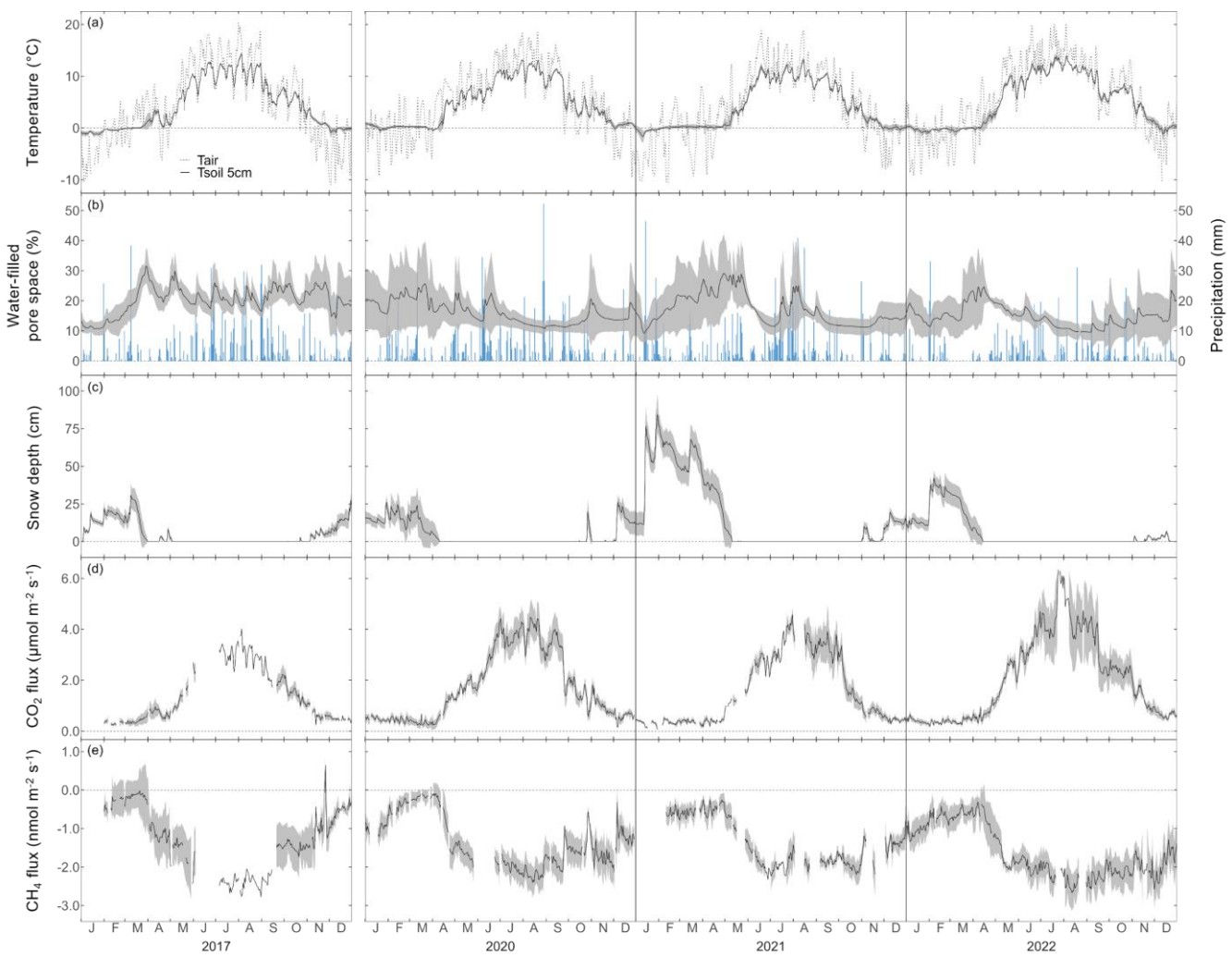

**Fig. 1: Daily mean a) air temperature and soil temperature at 5 cm depth, b) water-filled pore space at 5 cm depth (left axis) and daily sum of precipitation (right axis), c) snow depth, and daily mean forest-floor d) respiration fluxes (not gap-filled), and e) CH$_4$ fluxes (not gap-filled), for the years 2017, 2020, 2021, and 2022. Please note the gap in measurements between 2017 and 2020. Black lines show means over four chambers, grey bands show standard deviations among four chambers. All data shown were quality-**
280 **checked as described in the main text.**

The forest floor N$_2$O fluxes ranged between -100 and 200 nmol m$^{-2}$ h$^{-1}$ but were mostly between 0 and 30 nmol m$^{-2}$ h$^{-1}$, with a mean over both years of 18.9±22.5 nmol N$_2$O m$^{-2}$ h$^{-1}$ (measured with automatic chambers and laser spectrometer; Fig. 2a, b, c). Winter fluxes (November to April) were generally higher and showed higher variability compared to the rest of the year.
N$_2$O fluxes were within the calculated flux detection limit (29.1 nmol N$_2$O m$^{-2}$ h$^{-1}$) over a large part of the measurement period. N$_2$O fluxes measured manually with eight static chambers in October 2023 were low (mean ± SD = 2.9±31.1 nmol m$^{-2}$ h$^{-1}$,

Fig. 2d) and agreed very well with the fluxes measured using the automatic chambers (mean in October: $10.2\pm14.7$ nmol m$^{-2}$ h$^{-1}$). Both chamber measurements showed occasional N$_2$O uptake.

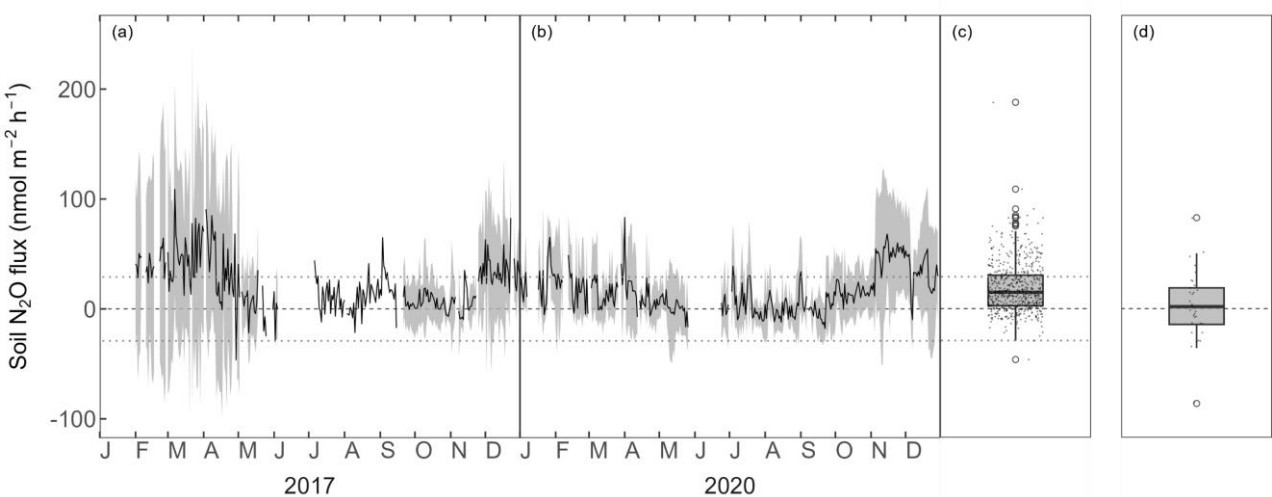

**Fig. 2: Forest-floor N$_2$O fluxes (nmol m$^{-2}$ h$^{-1}$) for the years a) 2017 and b) 2020. Black lines show means over four chambers, grey bands show standard deviations among four chambers. Boxplots show distribution of c) mean N$_2$O fluxes from four automatic chambers, and d) N$_2$O fluxes from static chamber measurements. The dotted lines depict the minimum flux (29.1 nmol N$_2$O m$^{-2}$ h$^{-1}$) which could be detected by the Dual Quantum Cascade Laser spectrometer.**

**3.2 Driver analyses with random forest models**

The RF models captured the temporal dynamics and absolute magnitudes of the observed forest-floor respiration and CH$_4$ fluxes very well, with $R^2$ values of 0.95 and 0.87, respectively (relationships of observed vs. predicted fluxes from test datasets), and RSME of 0.32 µmol m$^{-2}$ s$^{-1}$ and 0.27 nmol m$^{-2}$ s$^{-1}$, respectively (Fig. A.5). The seasonal RF models for forest-floor respiration fluxes yielded high $R^2$ values of 0.94, 0.73, 0.90 and 0.63 for spring, summer, autumn and winter, respectively

(Tab. A.3). Similarly, forest-floor CH$_4$ fluxes during spring, summer, autumn and winter were predicted well, with $R^2$ values of 0.80, 0.76, 0.72 and 0.73, respectively. Thus, the RF model performance was very good, also when shorter time periods were considered.

Forest-floor respiration fluxes combined for all four years and seasons were predominantly driven by T$_{soil}$ at 5 cm depth: T$_{soil}$ at the time of the flux measurements was the most important driver, but also T$_{soil}$ with a four-day (second most important) and

305 with a one-day lead were relevant (Fig. 3). Furthermore, WFPS at 5 cm with a four-day lead played an important role. As expected, higher T$_{soil}$ lead to higher respiration, while higher WFPS reduced forest-floor respiration. Drivers enhancing canopy photosynthesis, i.e., LAI or PPFD, did not play any role for forest-floor respiration. Separating forest-floor respiration fluxes into seasonal fluxes resulted in a clear distinction of drivers in winter compared to the other seasons (Fig. 3). Winter respiration

fluxes were mainly driven by snow depth (most important driver), leading to lower respiration fluxes with higher snow depths, while $T_{soil}$ played a smaller role. As for the overall fluxes, summer forest-floor respiration fluxes were mainly driven by $T_{soil}$, increasing with $T_{soil}$. Also total N stocks were highly relevant in summer, with higher total N stock leading to lower respiration fluxes, much in contrast to the fluxes during spring and fall (Fig. 3).

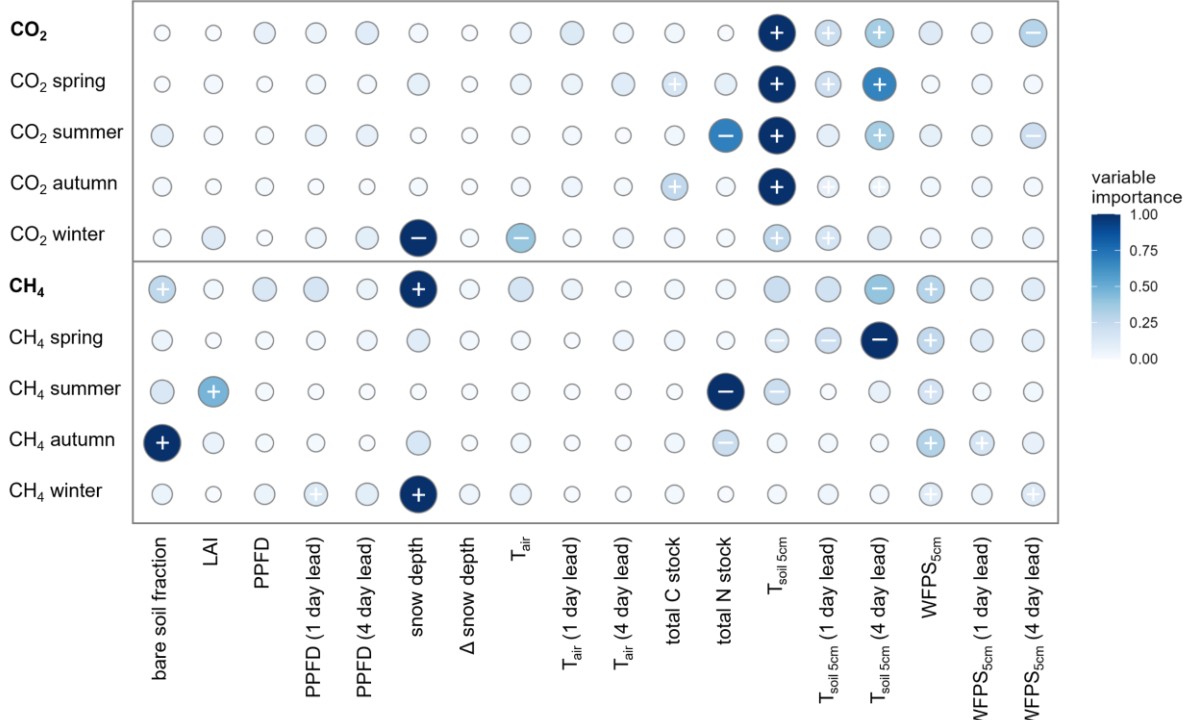

Fig. 3: Relative variable importance (rescaled to 0–1) according to the random forest driver analysis for forest-floor respiration (top; CO₂) and CH₄ (bottom) fluxes (not gap-filled; shown for the entire year, and per season). The direction of the effect of each predictor variable on the fluxes is shown by + (positive correlation) and – (negative correlation) signs, i.e., + indicates increased forest-floor respiration or decreased CH₄ uptake (i.e., increased CH₄ emissions). Signs are given for the four most important predictors which were investigated using partial dependence plots. See Materials and Methods for variable abbreviations.

The RF analysis showed that forest-floor CH₄ fluxes combined for all four years and seasons were mainly driven by the snow depth, with higher snow depths leading to more positive CH₄ fluxes and thus less CH₄ uptake (Fig. 3). Furthermore, the four-day lead of $T_{soil}$ at 5 cm impacted the fluxes negatively, leading to increased CH₄ uptake, while WFPS at 5 cm and the bare soil fraction inside the chamber lead to strongly decreased CH₄ uptake. We found that the drivers of the forest-floor CH₄ fluxes changed profoundly among seasons. Spring CH₄ fluxes were mainly temperature-driven (higher temperatures leading to more CH₄ uptake). In summer, forest-floor CH₄ fluxes were mainly driven by total N stocks (higher N stocks leading to more

negative $CH_4$ fluxes and thus higher uptake) and by LAI (higher LAI leading to more positive $CH_4$ fluxes and thus lower uptake), reflecting spatial variability among chambers. In addition, $CH_4$ fluxes were influenced by an interaction of several drivers such as $T_{soil}$ (higher $T_{soil}$ leading to higher uptake) and WFPS (higher WFPS leading to lower uptake). For autumn $CH_4$ fluxes, bare soil fraction was the most important driver (more bare soil – and thus smaller moss cover (Tab. A.2) – leading to more positive $CH_4$ fluxes and thus less $CH_4$ uptake), but also WFPS played an important role. Winter $CH_4$ fluxes responded mainly to snow depth, with higher snow depth leading to less $CH_4$ uptake (Fig. 3). Closer investigation of the relationship between the two most important drivers (snow depth and the four-day lead of $T_{soil}$) with daily $CH_4$ uptake over the entire year revealed that the temperature dependence of the $CH_4$ fluxes disappeared when snow was present (Fig. 4a). We found a significant logarithmic relationship between $CH_4$ uptake and snow depths, showing a decrease in $CH_4$ uptake with increasing snow depth (Fig. 4b). Additionally, observations of $CH_4$ release were mainly attributed to snow covered periods (85 % of positive $CH_4$ fluxes). Furthermore, the Spearman correlation coefficient between the $CH_4$ fluxes in the months October to May and snow depth was high with $r = 0.59$.

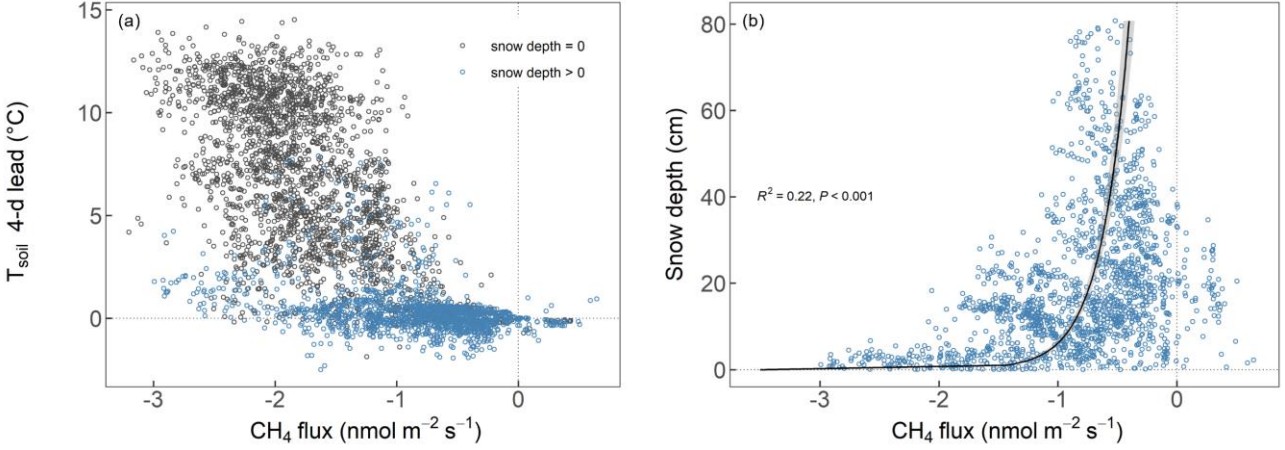

**Fig. 4: Relationship between forest-floor $CH_4$ fluxes (nmol $m^{-2}$ $s^{-1}$, daily means per chamber) and a) 4-day lead of soil temperature at 5 cm depth (°C) and b) snow depth (cm) with the black line showing the fitted logarithmic curve.**

### 3.3 Forest-floor GHG budgets

The forest floor of this subalpine spruce forest was a source of $CO_2$, a net sink of $CH_4$, and a close to zero $N_2O$ source for all years of the study (averaged over all four chambers; Tab. 1). Mean annual budgets were $2336\pm200$ g $CO_2$ $m^{-2}$ $yr^{-1}$ for forest-floor respiration, $0.007\pm0.009$ g $N_2O$ $m^{-2}$ $yr^{-1}$ for $N_2O$ emissions (two-year mean), and $-0.71\pm0.06$ g $CH_4$ $m^{-2}$ $yr^{-1}$ for $CH_4$ fluxes. The annual forest-floor respiration budgets were mainly determined by summer and early autumn fluxes (i.e., June to September). The interannual variability (SD) of forest-floor respiration budgets was approx. 200 g $CO_2$ $m^{-2}$ $yr^{-1}$ (8.6 %) during

the four years of the study, with 2017 and 2021 showing smaller and 2022 the highest emissions. The annual forest-floor respiration budgets calculated with the $Q_{10}$ modeled data ($2422\pm21$ g $CO_2$ $m^{-2}$ $yr^{-1}$; Tab. 1, Fig. A.4) agreed well with the forest-floor respiration budgets based on the gap-filled fluxes using RF, also showing highest fluxes in 2022. A similar interannual variability (SD) as for the respiration budgets was found for the $CH_4$ budgets, with 8.5 % (0.06 g $CH_4$ $m^{-2}$ $yr^{-1}$). Comparing the magnitudes of all three GHG fluxes (in $CO_2$-eq) clearly showed that the respiration budget dominated the forest floor GHG budget of the spruce forest. The forest floor $CH_4$ uptake ($-19.1\pm1.8$ g $CO_2$-eq $m^{-2}$ $yr^{-1}$) was about two orders of magnitude smaller than the respiration fluxes ($2336\pm200$ g $CO_2$-eq $m^{-2}$ $yr^{-1}$), while the annual forest-floor $N_2O$ emissions accounted for only $1.99\pm2.37$ g $CO_2$-eq $m^{-2}$ $yr^{-1}$, representing 0.09% of the annual forest-floor GHG budget.

**Tab. 1: Mean annual GHG budgets (±standard deviation (SD) over four chambers) of forest-floor respiration and $CH_4$ fluxes (using gap-filled data) and $N_2O$ fluxes (mean of two years of measurements). The $Q_{10}$ budget was calculated with Eq. 3 ($Q_{10}$ and $R_{ref}$ estimates were 4.8 and 3.16, respectively; overall $R^2$ was 0.86).**

| Year | Forest-floor respiration budget | | Forest-floor $CH_4$ budget | | Forest-floor $N_2O$ budget | | Net GHG budget based on RF |
|---|---|---|---|---|---|---|---|
| | based on RF (g $CO_2$ $m^{-2}$ $yr^{-1}$) | based on $Q_{10}$ (g $CO_2$ $m^{-2}$ $yr^{-1}$) | (g $CO_2$-eq $m^{-2}$ $yr^{-1}$) | (g $CH_4$ $m^{-2}$ $yr^{-1}$) | (g $CO_2$-eq $m^{-2}$ $yr^{-1}$) | (g $N_2O$ $m^{-2}$ $yr^{-1}$) | (g $CO_2$-eq $m^{-2}$ $yr^{-1}$) |
| 2017 | 2139±334 | 2407±28 | -17.1±3.6 | -0.63±0.13 | 2.36±2.69 | 0.008±0.010 | 2124±334 |
| 2020 | 2338±324 | 2390±54 | -18.7±3.3 | -0.69±0.12 | 1.66±2.00 | 0.006±0.007 | 2321±324 |
| 2021 | 2138±275 | 2204±40 | -18.3±2.7 | -0.68±0.10 | - | - | 2120±275 |
| 2022 | 2730±589 | 2687±40 | -22.2±4.4 | -0.82±0.16 | - | - | 2708±579 |
| Overall | 2336±200 | 2422±21 | -19.1±1.8 | -0.71±0.06 | 1.99±2.37 | 0.007±0.009 | 2319±200 |

The year 2022 can be considered an exceptional year, both in terms of annual forest-floor respiration and $CH_4$ fluxes (Tab. 1), but also in terms of temporal development (Fig. 5a). For $CO_2$, there were not only higher respiration rates in summer, but also a faster increase in respiration rates already in mid-April and sustained higher rates until later in the year (Fig. 5a). The exceptionally high forest-floor respiration fluxes (2022 forest-floor respiration budget falls outside the 95% confidence interval = ±1.96SD, i.e., for the forest-floor respiration budget: ± 392 g $CO_2$ $m^{-2}$ $yr^{-1}$) coincided with the higher-than-usual $T_{soil}$ (annual mean $T_{soil}$ of 2022 falls outside the 95% confidence interval) which was the main driver of spring, summer, and autumn forest-floor respiration fluxes. For $CH_4$, we observed a higher annual $CH_4$ uptake in 2022 compared to other years (Tab. 1), mainly due to higher uptake rates in summer as well as still high uptake rates in autumn and early winter (Fig. 5b). Apart from higher $T_{soil}$ driving the higher summer $CH_4$ uptake, this was mainly connected to comparably low soil moisture in autumn 2022 and the low snow depths in November and December 2022.

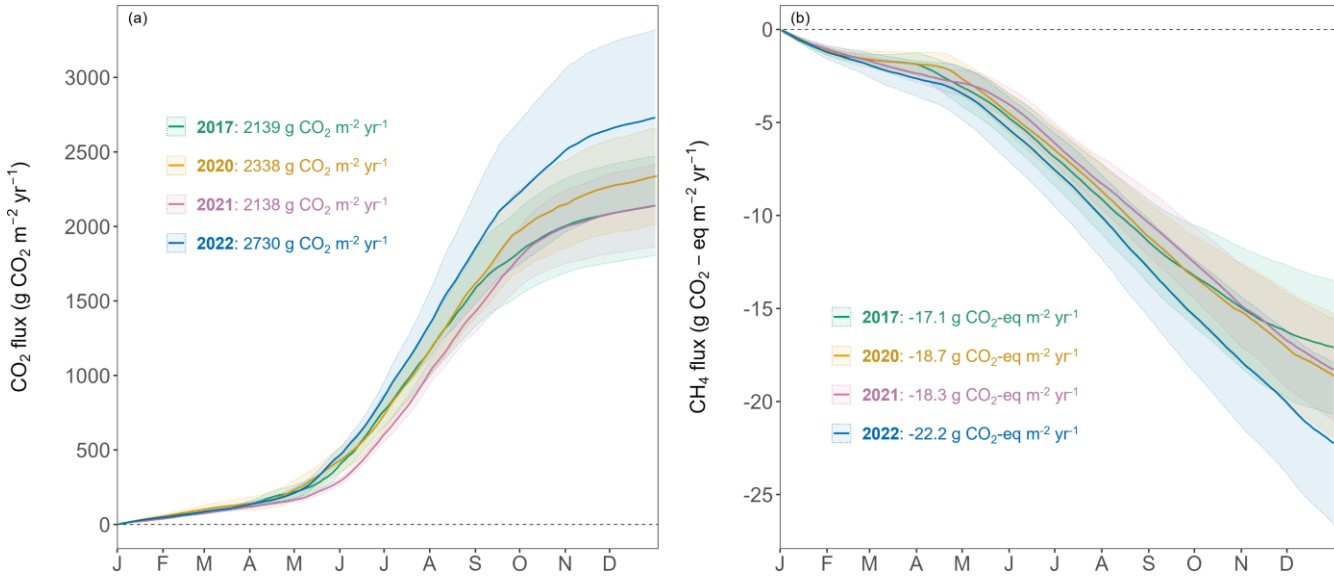

**Fig. 5: Cumulative forest-floor (a) respiration (g CO₂ m⁻² yr⁻¹) and (b) CH₄ (g CO₂-eq m⁻² yr⁻¹) fluxes over four years. Lines show**
**means of all four chambers; colored bands represent standard deviations among four chambers.**

## 4 Discussion

### 4.1 Interannual variability in forest-floor GHG fluxes

Over the four-year measurement period (2017 and 2020–2022), we collected high-resolution, reliable forest-floor GHG flux measurements for four very distinct years allowing comprehensive year-round analyses (two years for $N_2O$). Notably, 2022 emerged as the warmest year ever recorded at the Davos site so far, coinciding with remarkably low precipitation and WFPS levels. The forest-floor respiration in 2022 exceeded those in the other three years by approximately 20 %. Concurrently, we observed the highest forest-floor $CH_4$ uptake in 2022. It is well known that temperature is a major driver for any respiratory process (Davidson et al., 2006; Amthor, 2000). Also our RF driver analysis revealed that soil temperature was the main driver for forest-floor respiration fluxes, while no soil water limitation existed at this high elevation forest site during the study period. Anjileli et al. (2021) reported that at multiple sites across the contiguous United States even during extreme heat events, soil respiration increased by approximately 25 % compared to average conditions, emphasizing the dominating influence of temperature on respiration also under extreme dry conditions for those sites. Additionally, Borken et al. (2006) indicated that droughts can enhance the soil $CH_4$ sink in temperate forests. In contrast, the year 2021 was the coldest year among the four years we investigated, with an annual mean $T_{air}$ of 3.9 °C, mainly driven by below-average spring temperatures. This was reflected clearly in the forest-floor fluxes with below-average respiration rates (approximately 30 % lower compared to the four-year mean) and below-average $CH_4$ uptake in spring 2021 (approximately 20 % lower compared to the four-year mean).

Moreover, 2021 was an exceptional year in terms of snow depth, a relevant driver identified in this study (snow depth in winter and spring exceeded the four-year average by 87 % and 145 %, respectively).

While the year 2020 was also characterized by warm weather, its summer temperatures were less extreme than in 2022. Our findings revealed that the forest-floor respiration in 2020 did not reach the levels observed in 2022, supporting our driver analyses, clearly indicating that the exceptionally high summer temperatures experienced in 2022 were the primary driving force behind the 2022 annual forest-floor respiration fluxes. The RF models for 2022 resulted in slightly lower forest-floor respiration than measured, suggesting that no overfitting had occurred (Fig. A.6, A.7). Moreover, these results highlighted the critical role played by extreme summer temperatures in shaping the C dynamics of this subalpine spruce ecosystem and underscored the significance of understanding their implications for future C budgets, potentially reducing the overall C sink capacity observed so far in this forest (Zielis et al., 2014).

We measured very low forest-floor $N_2O$ fluxes which agreed well between the two measurement techniques used (automatic chambers and laser spectroscopy vs. static chambers and gas chromatography). Due to soil aeration and soil moisture conditions at our site, we assumed that nitrification and not denitrification was the main process responsible for the $N_2O$ emissions measured (Papen and Butterbach-Bahl, 1999; Butterbach-Bahl et al., 2013). At our site, N supply to plants and microorganisms is limited. Foliage N concentrations indicate N limitation for spruce (foliar N concentration are about 1 % in 0- and 1-yr-old needles as opposed to the optimum range of N content in needles between 1.5 and 2.3 %; Thimonier et al., 2010; Ingestad, 1959). Furthermore, N concentrations in the soil are low (1.4% in the organic layer, 0.4% in 10–20 cm depth; Jörg, 2008). N deposition at the site (about 10 kg N ha$^{-1}$ yr$^{-1}$; Thimonier et al., 2019; Gharun et al., 2021) corresponds to the lower level of critical N loads for forests in Switzerland (Hettelingh et al., 2017), well below the N deposition negatively related to basal area increments for spruce (20–22 kg N ha$^{-1}$ year$^{-1}$; Braun et al., 2017) or that with the highest positive effect on net ecosystem productivity, i.e., the C sink, of forests across Europe (22 kg N ha$^{-1}$ yr$^{-1}$; Wang et al., 2022). Thus, our site can clearly be considered rather low in N, which could be used for microbial transformations like nitrification, competing with plant uptake (Schulze, 2000), therefore, low soil $N_2O$ fluxes were to be expected. The observations of occasional low $N_2O$ uptake measured with static and automatic chambers are in line with Goldberg and Gebauer (2009) who observed $N_2O$ uptake in a German spruce forest. Microbial processes in forest soils can contribute to both uptake and release of $N_2O$, depending on the prevailing environmental conditions such as oxygen availability, soil moisture and microbial communities. Under anaerobic conditions, denitrification contributes to $N_2O$ release, while under aerobic conditions, $N_2O$ reduction to $N_2$ can dominate over $N_2O$ production, which results in observations of net $N_2O$ uptake by soils (Wen et al., 2017).

## 4.2 Drivers of forest-floor GHG fluxes

Forest-floor respiratory $CO_2$ and $CH_4$ fluxes were shown to have very distinct drivers across the different seasons. Consistent with our expectations, soil temperature predominantly controlled forest-floor respiration fluxes, thereby influencing the respiration budget at annual as well as seasonal scales (except winter season). In contrast, no effects of drivers known to

enhance canopy photosynthesis (i.e., LAI, PPFD) and thus below-ground allocation and soil respiration (Högberg et al., 2001) were observed on the forest-floor respiration fluxes for any time in this spruce forest, suggesting a strong direct control of environmental factors and only a weak or even lacking indirect control of canopy biology or structure. Drivers of forest-floor $CH_4$ fluxes were much more variable compared to those of forest-floor respiration fluxes, with winter $CH_4$ fluxes being affected by the same driver (snow depth) as the annual fluxes. The observations of lower $CH_4$ uptake during snow cover reflect the results of Heinzle et al., 2023 from a long-term soil warming experiment in a temperate forest. The findings that snow depth and WFPS (in autumn) were important drivers of forest-floor $CH_4$ fluxes supported the hypothesis proposed by Borken et al. (2006), who emphasized the role of factors influencing the diffusion rates of atmospheric $CH_4$ into the soil, such as SWC and snow cover, in determining $CH_4$ uptake in forest soils. Notably, previous studies had also reported a close relationship between $CH_4$ fluxes and seasonal changes in soil moisture (Ni and Groffman, 2018; Ueyama et al., 2015). However, our results indicated that in spring and summer, $T_{soil}$ rather than WFPS played a more important role in driving forest-floor $CH_4$ uptake. Additionally, we identified a notable influence of soil N on summer $CH_4$ fluxes, with higher N stocks, and thus most likely higher N mineralization during the summer months, corresponding to enhanced $CH_4$ uptake. This aligned with previous findings in forest ecosystems, where soil mineral N had been shown to stimulate $CH_4$ oxidation (Goldman et al., 1995; Martinson et al., 2021). Moreover, we found a positive correlation between bare soil fraction and forest-floor $CH_4$ uptake, i.e., more bare soil and thus lower moss cover leading to lower forest-floor $CH_4$ uptake. This is in line with findings that *Sphagnum* mosses can promote $CH_4$ oxidation (Basiliko et al., 2004). Also, for forest-floor $CH_4$ fluxes, hardly any effect of tree canopy biology was detected (except for summer). Thus, a strong direct control of environmental factors on both forest-floor respiration and $CH_4$ fluxes was observed, increasing the vulnerability of the forest C sink with future climate change (IPCC, 2021).

### 4.3 Forest-floor GHG budgets

The overall forest-floor GHG budget showed a total emission of $2319\pm200$ g $CO_2$-eq $m^{-2}$ $yr^{-1}$, dominated by the annual forest-floor respiration budget ($2336\pm200$ g $CO_2$ $m^{-2}$ $yr^{-1}$), which was within the range of studies conducted in temperate, subalpine or boreal forests which we considered comparable to our site ($1070$–$2906$ g $CO_2$ $m^{-2}$ $yr^{-1}$; Gaumont-Guay et al., 2014; Groffman et al., 2006; Schindlbacher et al., 2007, 2014; Wang et al., 2013; Xu et al., 2015). Also our estimate of annual $CH_4$ budget at the site ($-0.71\pm0.06$ g $CH_4$ $m^{-2}$ $yr^{-1}$) fell within the range of $-1.6$ to $-0.18$ g $CH_4$ $m^{-2}$ $yr^{-1}$ observed in other forest studies (Borken et al., 2006; Luo et al., 2013; Ueyama et al., 2015; Yu et al., 2017), offsetting a mere 0.8 % of forest-floor respiration. Our estimate of the annual $N_2O$ budget of $0.0073$ g $N_2O$ $m^{-2}$ $yr^{-1}$ agreed well with previous studies (Rütting et al., 2021; Ullah et al., 2009). For instance, a study conducted in a boreal spruce forest with low N deposition rates (about 5 kg N $ha^{-1}$ $yr^{-1}$) reported very low mean $N_2O$ fluxes of around $0.0077$ g $N_2O$ $m^{-2}$ $yr^{-1}$ (Rütting et al., 2021). Higher soil $N_2O$ emissions ($0.08$ g $N_2O$ $m^2$ $yr^{-1}$) have been observed in a temperate forest with higher N availability (N deposition rates 18 kg N $ha^{-1}$, N stocks in litter layer and mineral soil ~15 t $ha^{-1}$; Heinzle et al., 2023). Winter fluxes contributed a large fraction to the overall $CH_4$ budget (14.4–18.4 %), but played a less important role for the forest-floor respiration budget (6.0–7.3 %), similar to the $CO_2$

contribution in other mid latitude and temperate ecosystems (5.5–8.9 %; Gao et al., 2018; Wang et al., 2013) but smaller than some high latitude and other subalpine ecosystems (12–20 %; Kim et al., 2017; Schindlbacher et al., 2007; Xu et al., 2015).

To date, only a few studies have examined soil or forest-floor GHG fluxes in subalpine, temperate, or boreal forests, measuring $CO_2$, $CH_4$ *and* $N_2O$ fluxes in parallel (Tab. 2). Tab. 2 includes studies examining fluxes from both the forest floor (soil and ground vegetation) and the soil. However, their comparability is constrained as forest-floor flux measurements encompass both soil respiration (including heterotrophic and root respiration) and autotrophic respiration from forest-floor plants, whereas soil flux measurements specifically capture soil respiration (Barba et al., 2018). It is noteworthy that the integration of year-

round and temporally highly resolved measurements remains rather uncommon; to our knowledge, only two other studies with year-round measurements of $CO_2$, $CH_4$ *and* $N_2O$ exist apart from the current study (Luo et al., 2011; Pilegaard et al., 2003). On the one hand, previous studies frequently measured fluxes for only a limited period of the year, often excluding the dormant season. On the other hand, many of the studies adopted a weekly to monthly measurement frequency, potentially missing the full range of flux magnitudes. If year-round measurements of forest-floor respiration are not feasible, using $Q_{10}$ models might

be a viable option, as long as the annual temperature range is being well covered, as seen in the agreement between our respiration budget based on gap-filled continuous measurements and the $Q_{10}$ respiration budget. However, although $T_{soil}$ was identified as the primary driver of forest-floor respiration, it was not the only driver. We argue that $Q_{10}$ models are not able to capture extreme respiration fluxes which might be caused by more drivers than temperature alone. Many studies have shown that $Q_{10}$ models do not reproduce measured fluxes well when additional drivers impact the fluxes, for instance when soil

moisture, frost, or carbohydrate limitations come into play (e.g., Ruehr et al., 2010; Reichstein et al., 2013; Mitra et al., 2019). In contrast, our high-resolution dataset coupled with machine learning offered a more comprehensive approach, which included multiple environmental variables and at the same time was able to consider chamber-specific characteristics, and thus was able to capture the extreme fluxes we observed in summer 2022. Thus, we think that the reliability of the RF budget is higher than that of the $Q_{10}$ budget. Moreover, identifying important drivers for GHG fluxes is the more reliable, the longer and thus

typically the more frequent measurements were done. Additionally, to effectively capture the dynamic nature of soil and/or forest-floor fluxes, it is essential to use automatic chambers with high temporal resolution, preferentially opaque to exclusively quantify respiration. Therefore, we recommend continuous, year-round measurements to reliably estimate annual forest-floor GHG budgets, particularly when large seasonal variability of potential drivers is expected, or when the duration of the active period, i.e., start and end of the snow-free period, is highly variable like in high elevation or high latitude ecosystems.

Particularly with the anticipated impacts of future climate change (IPCC, 2021), duration of growing periods will change, and winter fluxes (or the lack thereof) will gain increasing importance (Xie et al., 2017).

**Tab. 2: Previously published studies investigating forest-floor or soil CO₂, CH₄ _and_ N₂O fluxes in parallel in temperate, subalpine, or boreal forests using automatic or static chambers. n.a. = not available.**

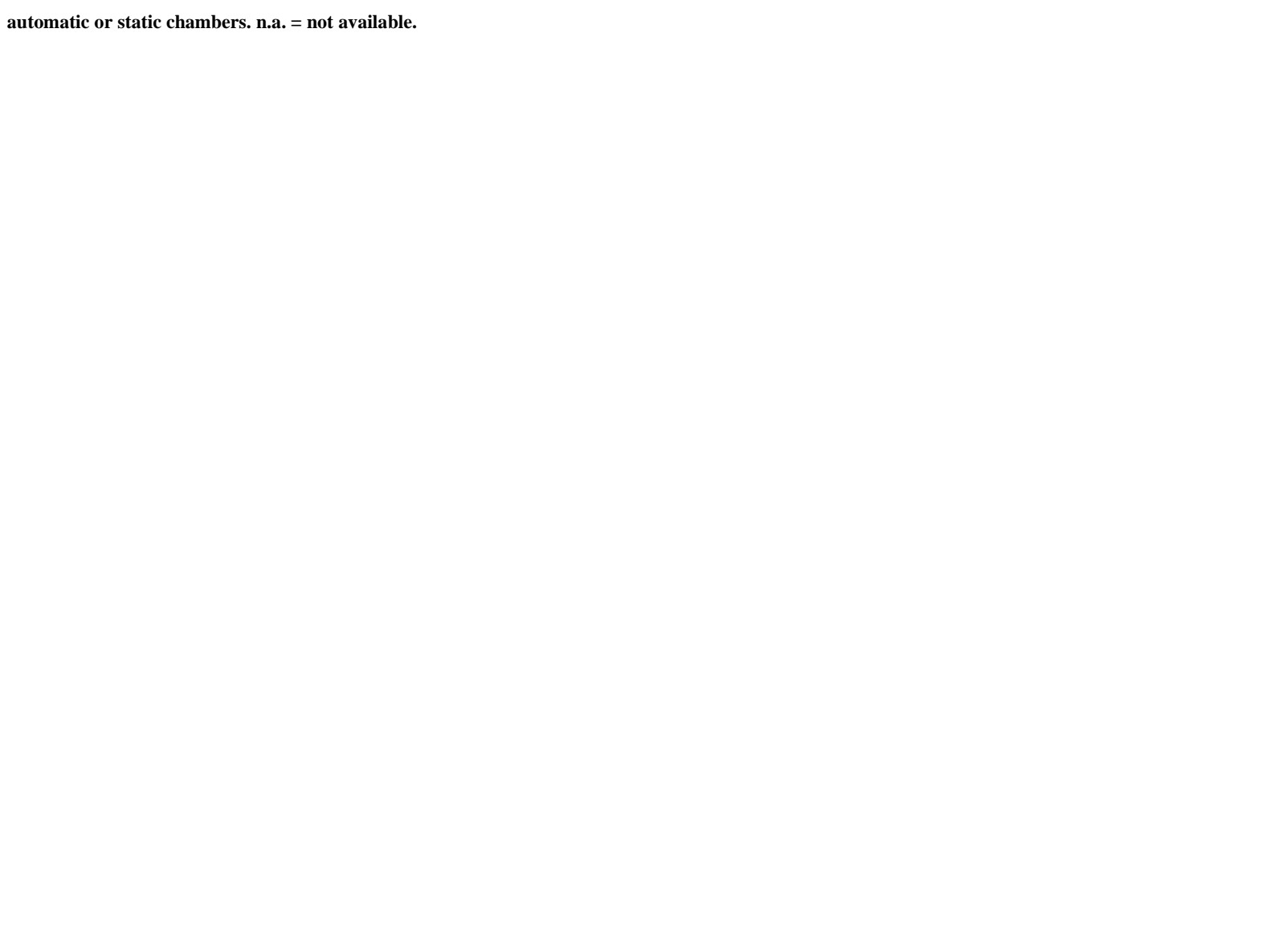

| Chamber method | Location | Forest type | Years | Duration | No. chambers | Frequency | Veg. in chambers | $CO_2$ flux rates ($\mu$g $CO_2$ m$^{-2}$ s$^{-1}$) | $CH_4$ flux rates (ng $CH_4$ m$^{-2}$ s$^{-1}$) | $N_2O$ flux rates (ng $N_2O$ m$^{-2}$ s$^{-1}$) | Reference |
|---|---|---|---|---|---|---|---|---|---|---|---|
| Automatic | 46.82° N 9.86° E | Subalpine (spruce) | 2017, 2020–2022 | Year-round | 4 | 3 h | Yes | 74.1±6.3 | -22.5±1.9 | 0.2±0.3 | This study |
| Automatic | 39.09° N 75.44° W | Temperate (mixed) | 2017 | Apr–Jul | 3 | 1 h | No | 362.6±24.2 | -10.6±1.0 | -2.0±0.5 | Barba et al., 2019 |
| Static | 43.23° N 3.20° W | Radiata pine, Douglas fir, beech | 2010–2011 | Year-round | 6 | Biweekly | Yes | 14.7±1.6 | 0.8±0.1 | 1.3±0.4 | Barrena et al., 2013 |
| Static | 37.07° N 119.19° W | Montane mixed-conifer (Mediter-ranean-type climate) | 2010–2012 | Snow free period | 24 | Weekly–monthly | n.a. | 51.7–63.3 | -9.6–-4.8 | -0.3–1.7 | Blankinship et al., 2018 |
| Static | 35.66° S 148.15° E | Temperate (eucalypt) | 2006 | 2 weeks in Nov | 10 | 4 h | No | 90.4±1.9 | -19.2±0.4 | 1.4±0.04 | Fest et al., 2009 |
| Static | 43.93° N 71.75° W | Northern hardwood (beech, maple, birch) | 1998–2000 | Year-round | 8 | Weekly–monthly | n.a. | 26.7–46.5 | -20.0–-7.9 | 2.0–7.0 | Groffman et al., 2006 |
| Static | 42.40° N 128.10° E | Broad-leaved Korean pine mixed | 2019 | Mar–Oct | 8 | Twice a week–twice a month | n.a. | 241.0±114.9 | -35.9±12.5 | 9.7±6.2 | Guo et al., 2020 |
| Static | 48.09° N 16.01° E | Temperate (beech) | 1997 | Apr–Nov | 8 | Biweekly | Yes | 53.0–57.8 | -5.6–-3.2 | 11.9–30.5 | Hahn et al., 2000 |
| Static | 43.83° N 74.87° W | Temperate (mixed) | 2008 | May–Jul | 15 | Biweekly | Yes | 10.2–101.8 | -16.7–42.1 | -1.2–2.8 | Hopfens-perger et al., 2009 |
| Static | 47.03° N 8.72° E | subalpine (spruce) | 2007–2012 | Year-round | 10 | Every 3 weeks | Yes | 48.8 | -8.0–13.4 | -1.2–2.9 | Krause et al., 2013 |
| Automatic ($CO_2$), static ($CH_4$, $N_2O$) | 48.50° N 11.17° E | Temperate (spruce) | 1994–1997, 2000–2010 | Year-round | 5 | 1 h ($CO_2$), 2 h ($CH_4$, $N_2O$) | n.a. | 81.3–106.9 | -14.8–-3.8 | 1.0–14.9 | Luo et al., 2011 |
| Static | 46.67–47.93° N 91.75–92.52° W | Boreal-temperate (mixed) | 2013 | May–Oct | 48 | Monthly | Yes | 0.002–0.004 | -0.0014–-0.0003 | -10.3–10.3 | Martins et al., 2017 |
| Static | 33.30–33.47° N 108.35–108.65° E | Temperate–cold temperate (deciduous broad-leaved & coniferous) | 2012–2014 | Year-round | 60 | Weekly–monthly | Yes | 44.4–86.9 | -24.0–-3.8 | 5.9–11.2 | Pang et al., 2023 |

| Method | Coordinates | Forest type | Years | Season | N | Frequency | Annual | CO2 | CH4 | N2O | Reference |
|---|---|---|---|---|---|---|---|---|---|---|---|
| Automatic (CO$_2$), static (CH$_4$, N$_2$O) | 55.48° N 11.63° E | Temperate (beech) | 1998–1999, 2001 | Year-round | 5 (CO$_2$), 6 (CH$_4$, N$_2$O) | 2 h (CO$_2$), biweekly (CH$_4$, N$_2$O) | Yes | n.a. | n.a. | n.a. | Pilegaard et al., 2003 |
| Automatic | 45.20° N 68.74° W | Sub-boreal (spruce, hemlock) | 2013–2016 | May–Nov | 3-5 | 30 min | n.a. | n.a. | n.a. | n.a. | Richardson et al., 2019 |
| Concen-tration profiles | 41.33° N 106.33° W | Subalpine (spruce, fir) | 1991–1992 | Mar–May | 2 | Daily–biweekly | Yes | 18.3–31.6 | -0.0029–-0.0004 | 0.0003–0.0004 | Sommerfeld et al., 1993 |
| Static | 49.26–52.20° N 74.03–76.07° W | Boreal (black spruce, jack pine, aspen, alder) | 2007 | May–Oct | 48 | Monthly | Yes | 34.4–64.0 | -6.7–1.6 | 0.4–0.8 | Ullah et al., 2009 |
| Static | 57.13° N 14.75° E | Cold temperate (coniferous) | 1999–2002 | Year-round | 30 | Weekly–biweekly | Yes | 28.5–60.2 | 0.0–50.7 | 1.0–2.9 | Von Arnold et al., 2005 |
| Static | 53.28–53.50° N 122.10–122.45° E | Cold temperate continental monsoon | 2016–2018 | Year-round | 9 | Weekly–monthly | Yes | 2.2–180.8 | -15.9–9.0 | -1.1–8.6 | Wu et al., 2019 |

## 5 Conclusions

Forest-floor GHG fluxes, measured during multiple years with large opaque automatic chambers, were mainly driven by environmental factors, with only limited impacts of tree biology or structure. Particularly, in light of climate change-induced variations in the onset of the active growing season, growing season length, and winter conditions, we recommend to spatially expand the deployment of such chambers at research stations capable of year-round measurements, including periods with snow cover. Since our forest study site was very low in N supply and thus $N_2O$ fluxes were very low, it remains to be seen how large annual $N_2O$ emissions are from other forest sites with higher N supply and what drivers are most relevant there. As temperatures will continue to rise due to climate change, and warm and dry conditions such as in the recent summers are projected to become more frequent and more severe, we expect an increase in forest-floor respiration at the Davos spruce forest and similar subalpine or high latitude ecosystems. Similarly, anticipated milder winters with reduced snowfall, resulting in shorter snow cover duration and lower average snow depth, will likely contribute to enhanced forest-floor respiration and increased forest-floor $CH_4$ uptake in the future. Since respiratory $CO_2$ losses are typically much larger than $CH_4$ uptake rates, as at our site, we expect the forest floor to become a more substantial C source in the future, potentially modulating the overall C sink capacity of this type of forest.

*Data availability.* The data used in this study will be made publicly available from the ETH Research Collection (https://doi.org/10.3929/ethz-b-000619728).

*Author contributions.* NB designed the study; PM, LK and SB conducted the field work; SB and LK processed the data; LK performed the data analyses, developed the models, and wrote the manuscript draft; SB, MG, PM, IF and NB commented on the manuscript and contributed substantially to discussions and revisions.

*Competing interests.* The authors declare that they have no conflict of interest.

*Acknowledgements.* The authors thank our colleagues Lutz Merbold, Matti Barthel, Lukas Hörtnagl, Thomas Baur, Werner Eugster and Liliana Scapucci for their assistance in designing and setting up the chambers, conducting fieldwork, and providing helpful inputs during the flux processing and data interpretation. Their contributions have greatly contributed to the progress of this study.

*Financial support.* This research has been supported by the Swiss National Science Foundation (SNSF), in the projects ICOS-CH Phase 1, 2, 3 (Grant-N° 20FI21_148992, 20FI20_173691, 20F120_198227) and COCO (Grant-N° 200021_197357).

## A Appendix

**Tab. A.1: 10th percentiles of $R^2$ values from linear regressions used for flux calculations per gas, given separately for each chamber (FF1 to FF4) and growing and dormant periods. Percentiles were applied as quality thresholds.**

| Gas | Period | FF1 | FF2 | FF3 | FF4 |
|-----|--------|-----|-----|-----|-----|
| CO$_2$ | growing period | 0.97 | 0.98 | 0.98 | 0.99 |
| | dormant period | 0.35 | 0.48 | 0.47 | 0.68 |
| CH$_4$ | growing period | 0.92 | 0.96 | 0.92 | 0.93 |
| | dormant period | 0.41 | 0.26 | 0.21 | 0.61 |

**Tab. A.2: Site characteristics of the four chambers (FF1 to FF4). Annual means and standard deviations are shown for soil temperature (T$_{soil}$) and water filled pore space (WFPS) at 5 cm, mean and max snow depth, and days with snow cover. LAI, soil, and vegetation cover inside each chamber were determined in June 2022. Soil data (bulk density, pH, C and N stocks in the topsoil, i.e., litter, organic material layers, and 0–20 cm depth of mineral soil) were taken from Jörg (2008) and Saby et al. (2023).**

| Site characteristics | FF1 | FF2 | FF3 | FF4 | Mean |
|----------------------|-----|-----|-----|-----|------|
| T$_{soil}$ at 5cm (°C) | | | | | |
| 2017 | 4.44 ± 4.67 | 4.16 ± 4.84 | 4.29 ± 4.86 | 4.56 ± 4.52 | 4.36 ± 0.17 |
| 2020 | 4.66 ± 4.32 | 4.40 ± 4.46 | 4.46 ± 4.34 | 4.87 ± 4.15 | 4.60 ± 0.22 |
| 2021 | 4.18 ± 4.25 | 3.80 ± 4.48 | 3.74 ± 4.84 | 4.26 ± 4.20 | 3.99 ± 0.26 |
| 2022 | 5.15 ± 4.70 | 4.83 ± 4.96 | 4.70 ± 5.38 | 5.18 ± 4.61 | 4.97 ± 0.24 |
| WFPS at 5 cm (%) | | | | | |
| 2017 | 20.1 ± 5.09 | 17.2 ± 4.30 | 21.3 ± 6.82 | 22.5 ± 7.42 | 20.3 ± 2.27 |
| 2020 | 15.9 ± 2.88 | 15.5 ± 3.85 | 9.8 ± 0.69 | 23.9 ± 9.55 | 16.3 ± 5.79 |
| 2021 | 16.8 ± 3.88 | 14.5 ± 3.88 | 11.8 ± 4.45 | 25.0 ± 10.5 | 17.0 ± 5.70 |
| 2022 | 15.1 ± 4.19 | 12.7 ± 3.27 | 10.4 ± 3.56 | 21.3 ± 7.16 | 14.9 ± 4.70 |
| Max snow depth (cm) | | | | | |
| 2017 | 34.7 | 40.7 | 27.4 | 25.6 | 47.4 ± 24.5 |
| 2020 | 31.8 | 41.7 | 25.9 | 22.0 | 58.6 ± 30.0 |
| 2021 | 83.9 | 103.0 | 79.5 | 62.6 | 43.5 ± 25.0 |
| 2022 | 39.4 | 48.7 | 41.1 | 36.8 | 36.8 ± 18.3 |
| Mean snow depth (cm) | | | | | |
| 2017 | 5.8 ± 8.4 | 6.4 ± 9.8 | 4.5 ± 6.4 | 3.9 ± 6.2 | 5.1 ± 1.2 |
| 2020 | 4.3 ± 7.0 | 8.6 ± 12.1 | 4.2 ± 7.0 | 3.5 ± 6.0 | 5.2 ± 2.3 |
| 2021 | 17.6 ± 25.0 | 22.2 ± 29.5 | 14.6 ± 22.7 | 14.8 ± 21.0 | 17.3 ± 3.6 |
| 2022 | 5.1 ± 9.3 | 8.3 ± 14.1 | 6.1 ± 11.7 | 4.9 ± 9.5 | 6.1 ± 1.6 |
| Days with snow cover | | | | | |
| 2017 | 152 | 159 | 152 | 148 | 153 ± 5 |
| 2020 | 126 | 142 | 123 | 117 | 127 ± 11 |
| 2021 | 172 | 189 | 161 | 169 | 173 ± 12 |
| 2022 | 138 | 145 | 134 | 132 | 137 ± 6 |
| Leaf area index (LAI) | 2.9 | 4.2 | 3.1 | 2.9 | 3.3 ± 0.6 |
| Soil cover inside chamber (%) | | | | | |
| bare soil | 0 | 50 | 70 | 0 | 30 ± 36 |
| moss | 90 | 50 | 20 | 90 | 63 ± 34 |
| grass | 5 | 1 | 0 | 0 | 2 ± 2 |

| | | | | | |
|---|---|---|---|---|---|
| *Vaccinium* | 60 | 0 | 10 | 30 | $25 \pm 26$ |
| Bulk density at 5 cm of mineral soil (g cm$^{-3}$) | 0.27 | 0.35 | 0.32 | 0.35 | $0.32 \pm 0.04$ |
| pH | 2.8–3.1 | 3.0–3.4 | 2.8–3.1 | 3.0–3.4 | |
| C stock (t/ha) | 93.5 | 147.7 | 135.4 | 105.8 | $120.6 \pm 25.2$ |
| N stock (t/ha) | 3.54 | 5.74 | 4.47 | 3.52 | $4.32 \pm 1.05$ |

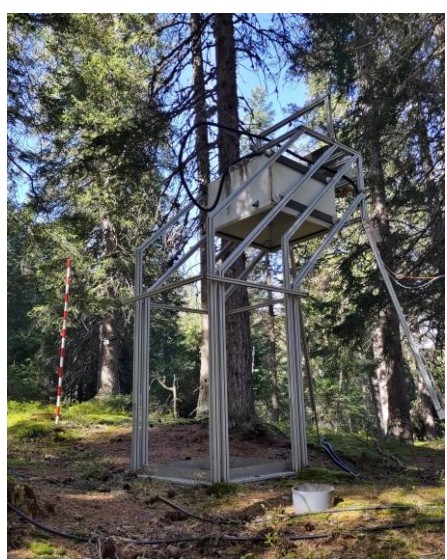

**Fig. A.1: Picture of one of the automatic chambers (at location FF3).**

**Tests for chamber biases**

We tested for chamber effects using SWC measurements from inside and outside FF1 and FF2 over all four years (Fig. A.2).
SWC was highly variable over time as well as in space (Fig. A.2a). SWC differences between inside and outside the chamber varied between +10% and -10% during the four years (Fig. A.2b). No clear trend was detectable over time. The average difference between inside and outside SWC over the four years was -2.9±5.8%. No significant differences in SWC inside vs. outside the chamber were detected during most of the year (exception: during winter, on average 5% lower SWC values inside compared to outside of the chamber). We found a high agreement in the dynamics of SWC inside and outside FF1 and FF2
($R^2$ values of 0.69 and 0.82, respectively). In terms of $T_{soil}$, we did not find any significant differences inside vs. outside the chambers over most of the year (Fig. A.3a). The differences were only significantly different from zero in the months December, February, and March when $T_{soil}$ inside the chambers was around 0.1-0.5 °C lower than outside the chambers (Fig. A.3b). At prevailing soil temperatures of around 0 °C in these months, such a difference in $T_{soil}$ has no effect on the magnitude of forest-floor respiration (Fig. A.4).


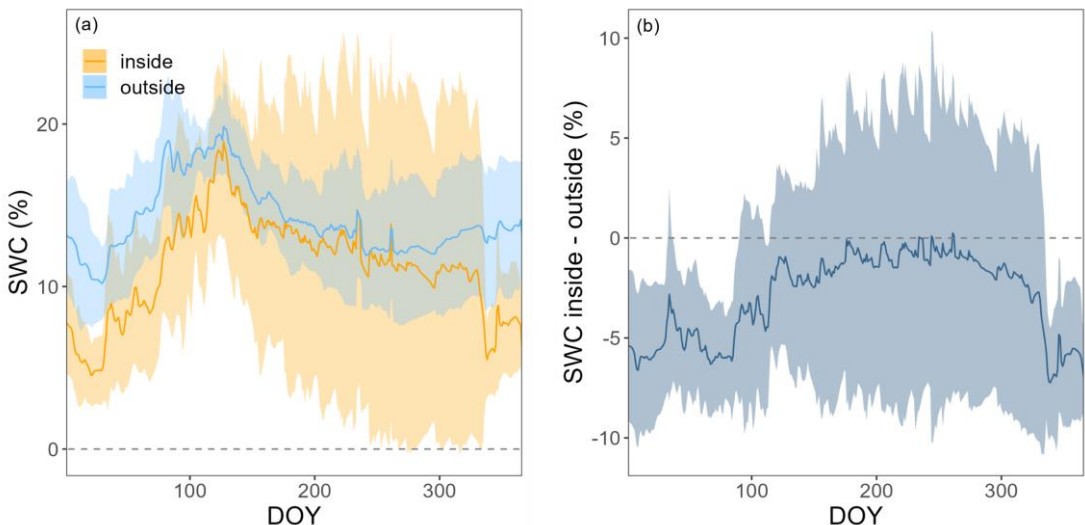

**Fig. A.2: a)** Soil water content (SWC) at 5 cm inside (orange) and outside (light blue) and **b)** the difference in SWC at 5 cm between inside and outside the chambers FF1 and FF2 over the course of a year. Lines show means, bands show standard deviations over all four years.

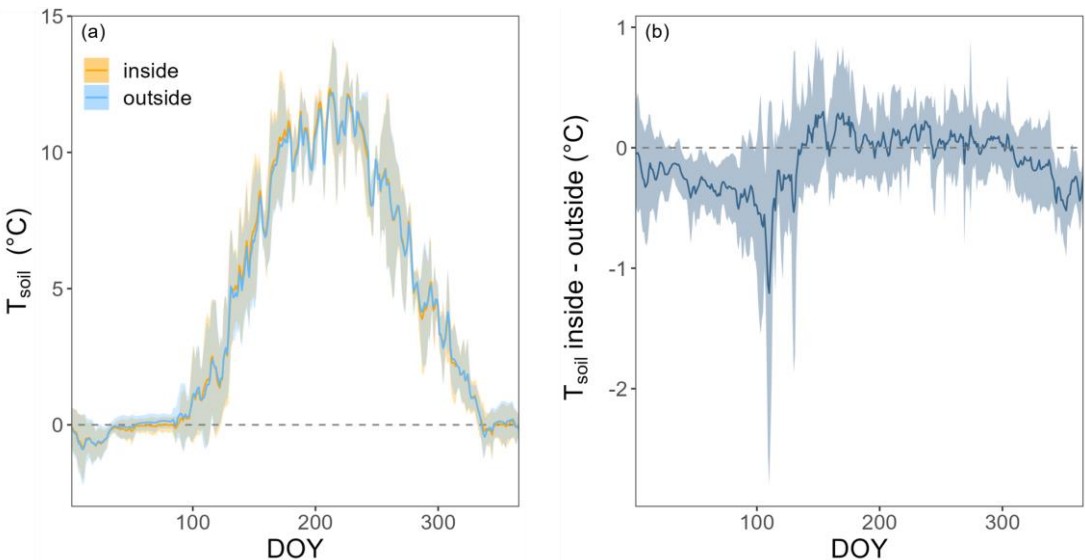

**Fig. A.3: a)** Soil temperatures ($T_{soil}$) at 5 cm inside (orange) and outside (light blue) and **b)** difference in $T_{soil}$ at 5 cm between inside and outside of chambers (FF1 to FF4) over the course of a year. Lines show means, bands show standard deviations over three years (2017, 2020 and 2021).

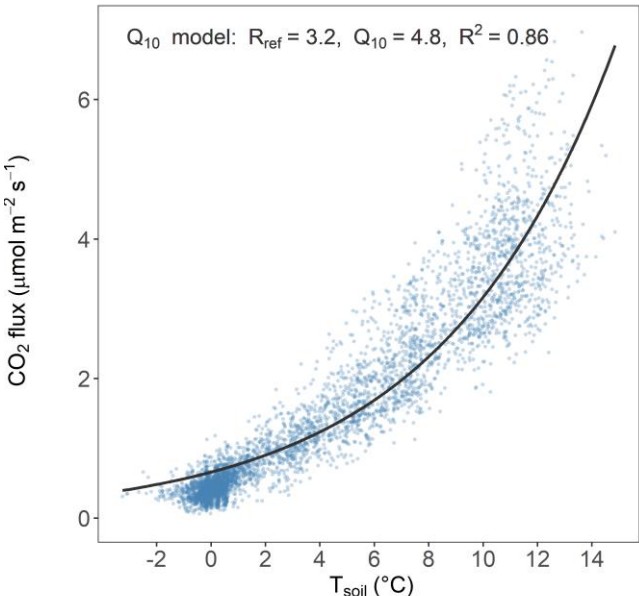

**Fig. A.4: $Q_{10}$ model showing the relationship of daily means of $T_{soil}$ at 5 cm and forest-floor respiration of all chambers and years.**

**Tab. A.3: Details of random forest models used for driver analysis and gap-filling for different time periods (entire year, separately for seasons). Number of observations used to train the models (training set), hyperparameters "mtry" and "ntree" as well as $R^2$**
**values for observed vs. predicted test data are given. "mtry" specifies how many variables were randomly sampled as candidates at each split, "ntree" indicates the number of trees.**

| Gas | Time period | No. observations in training set | mtry | ntree | test $R^2$ |
|---|---|---|---|---|---|
| $CO_2$ | entire year | 3111 | 10 | 2000 | 0.95 |
| | spring | 860 | 18 | 2000 | 0.94 |
| | summer | 623 | 14 | 2000 | 0.73 |
| | autumn | 836 | 14 | 2000 | 0.90 |
| | winter | 774 | 14 | 2000 | 0.63 |
| $CH_4$ | entire year | 2799 | 14 | 2000 | 0.87 |
| | spring | 825 | 18 | 2000 | 0.80 |
| | summer | 520 | 18 | 2000 | 0.76 |
| | autumn | 772 | 10 | 2000 | 0.72 |
| | winter | 674 | 10 | 2000 | 0.73 |

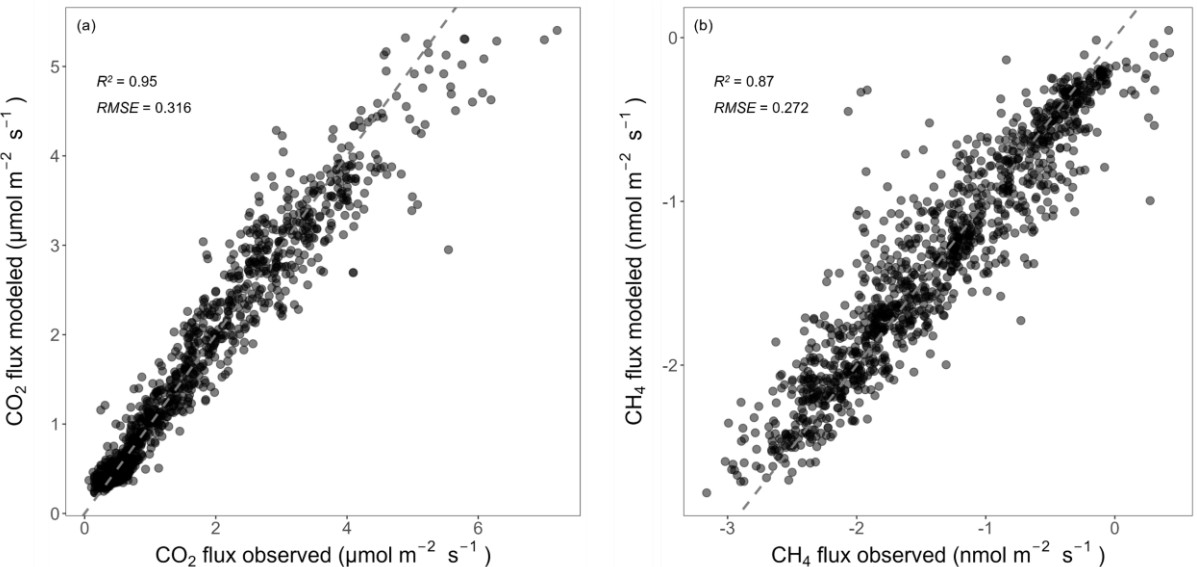

**Fig. A.5: Relationships between observed and predicted forest-floor (a) respiration and (b) CH₄ fluxes from the RF models used for gap filling (only showing test data). R² and RSME are given. Black dashed lines mark the 1:1 lines.**

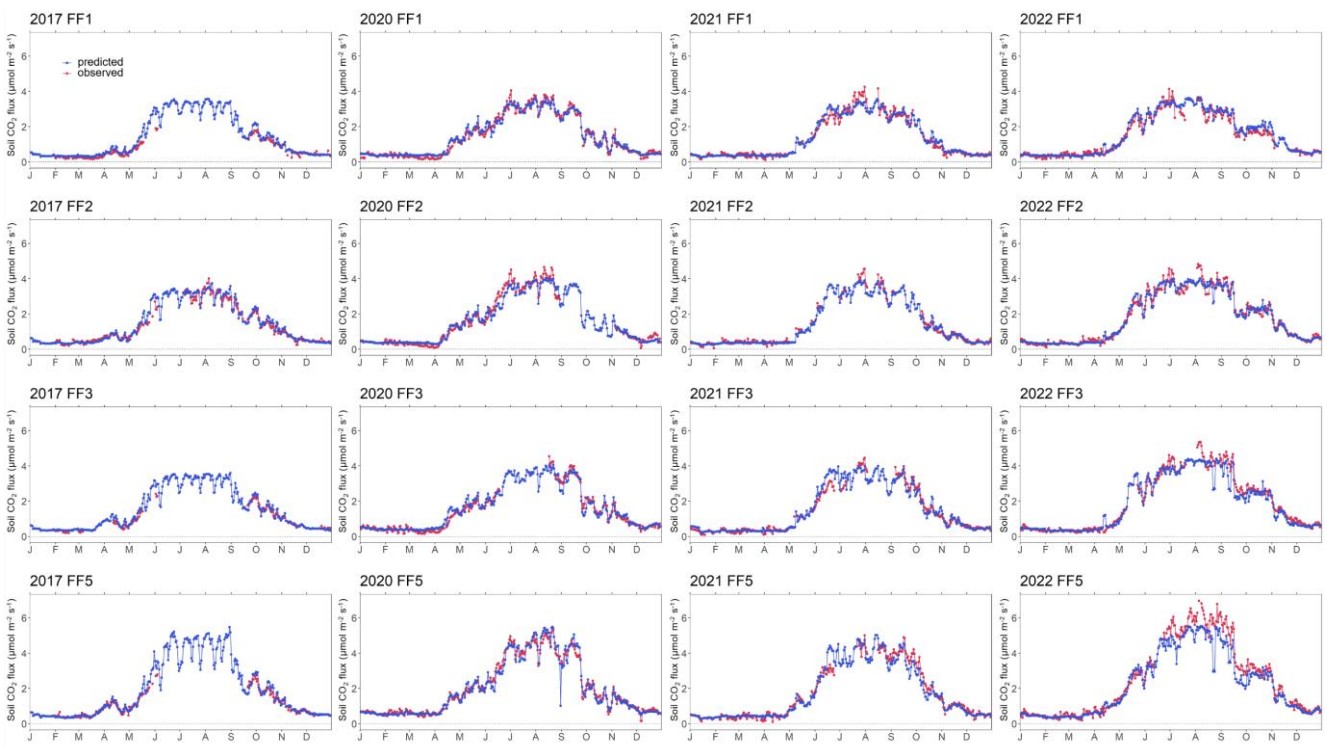

 Fig. A.6: Time series of observed and predicted (using random forest model) forest-floor respiration fluxes for four years (2017, 2020–2022) and four chambers (FF1 to FF4).

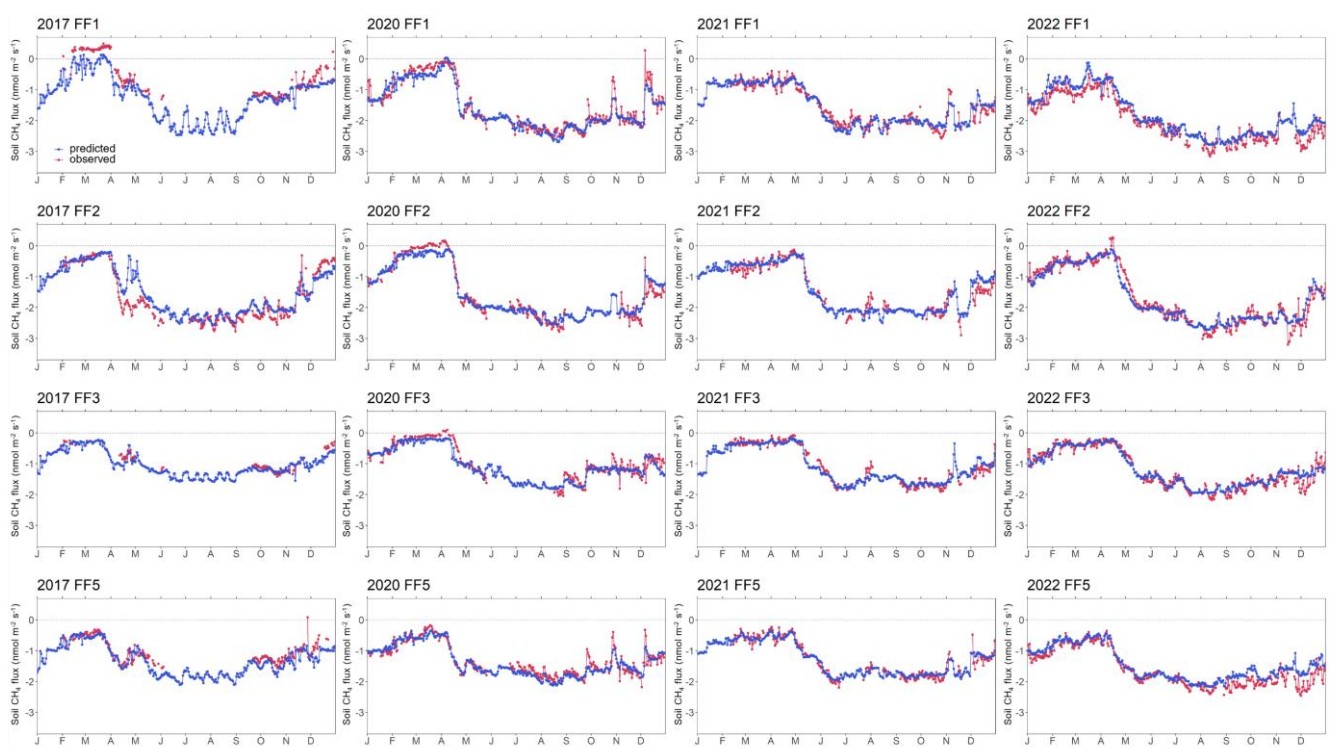

Fig. A.7: Time series of observed and predicted (using random forest model) forest-floor CH₄ fluxes for four years (2017, 2020–2022) and four chambers (FF1 to FF4).

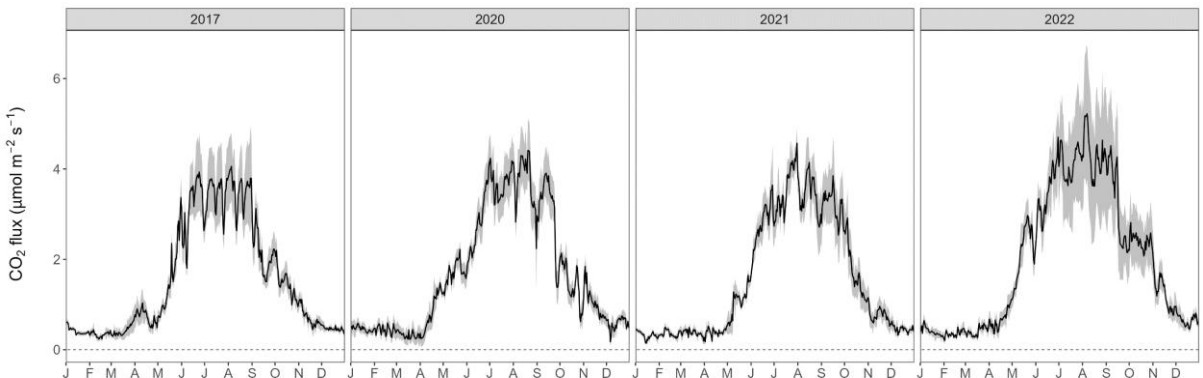

Fig. A.8: Gap-filled forest-floor respiration fluxes over four years (grey area: min-max among four chambers).

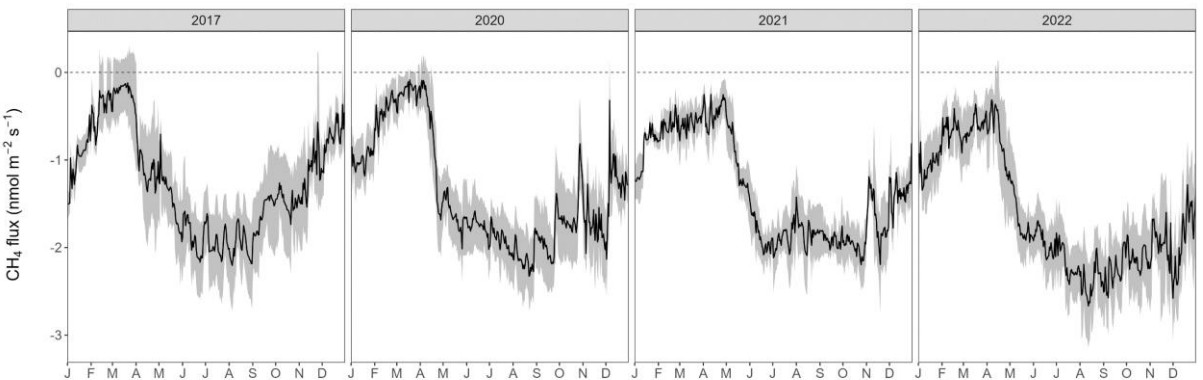

**Fig. A.9: Gap-filled forest-floor CH₄ fluxes over four years (grey area: min-max among four chambers).**

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
