# Peer review of "Forest-floor respiration, $N_2O$ , and $CH_4$ fluxes in a subalpine spruce forest: Drivers and annual budgets"

_EGUsphere, 2023_

## Author Comment (AC1)

**Discussion of "Forest-floor greenhouse gas fluxes in a subalpine spruce forest: Continuous multi-year measurements, drivers, and budgets"**

**Author Response to Referee 1 comments**

Krebs et al.

November 30, 2023

In the following, *reviewer comments are given italics*, author comments are given in normal font.

**1.    General comments**

*Luana Krebs and co-authors present an impressive soil GHG flux dataset from a subalpine forest. Multi-year datasets from such ecosystems are rather scarce and therefore definitely deserve publication. CO2, CH4 (and N2O) were measured in high temporal resolution year-round during 4 years and annual CO2equ budgets were calculated showing higher net emission during the warmest year 2022. I have some suggestions to improve the manuscript.*

Thanks for the comment and your suggestions.

*It should be considered removing N2O from the manuscript. Obviously N2O data is available only for 2 years and the extremely low fluxes are only shown in the supplements and not included in the budgets or any other calculations. It appears that the 180 sec chamber closure likely were not long enough for serious N2O flux calculations (extremely low 10th percentile R2 in Table.2) – as mentioned in the manuscript, using such a poor fit to calculate fluxes should be avoided. I don't know the actual cause. It is unlikely that the subalpine soil does not show any, or not measurable N2O emissions throughout the whole year. It seems more likely that there appeared problems with the laser. I don't operate one by myself but heard from colleagues that the real measurement accuracy in the field is not really the 0.06 ppb for N2O that is suggested. There might be drift or whatever else. How and how often was the laser calibrated? If you do not have 100% trust in the N2O data, I would take them completely out of the manuscript. Currently you state that there was no soil N2O efflux at this site – are you 100% sure?*

The suggestion to remove the $N_2O$ fluxes from the manuscript seems to be based on five reasons:
1) data available only for 2 years,
2) very low $N_2O$ fluxes,
3) 180 sec closure for flux measurements and low $R^2$ values,
4) unlikely that alpine soils show such small fluxes, and
5) problems with the laser (accuracy, drift).
In the following we want to address those five points.
1) Long-term $N_2O$ fluxes in forests are very rare, due to the difficulties measuring year-round. Thus, two years of $N_2O$ flux data are actually extraordinary, and there is no reason to delete such rare fluxes.

2) We agree that the $N_2O$ fluxes were low, and we did not use them for further driver analysis, one of the main objectives of the manuscript. This decision was made mainly because of the low magnitude of the fluxes and their irrelevance for the forest-floor GHG budget (using the mean $N_2O$ flux measured with the automatic chambers over the two years, 0.63 nmol $m^{-2}$ $h^{-1}$, we arrive at an annual budget of 0.066 g $CO_2$-eq $m^{-2}$ $yr^{-1}$ which represents 0.003% of the annual forest-floor GHG budget). However, we think that the result of the irrelevance of $N_2O$ fluxes for the annual forest-floor GHG budget of the forest does not diminish the importance of these fluxes.

3) In order to check for the validity of our chamber measurements, we have performed measurements using static chambers with the dimensions of d = 30 cm and h = 30 cm (Hutchinson and Mosier, 1981). We used eight static chambers, i.e., four chambers next to the automatic chambers, and four chambers placed randomly within the research area. Soil collars were installed two weeks prior to the first measurement campaign. Four rounds of sampling were done on two measurement days in October 2023 (n=32), when soil temperatures were between 5.5-10 °C, well above the long-term mean, and soil moisture values above 8%, favoring microbial activities. Three collars were irrigated between the first and second sampling round on the two days to simulate a heavy rainfall event, favoring denitrification. We left the chambers closed for 1 h and sampled the air in the headspace every 20 min. The fluxes that we measured with the eight static chambers were low (mean±SD = 2.9±31.1 nmol $m^{-2}$ $h^{-1}$). Furthermore, they agreed very well with the fluxes which we have measured using the automatic chambers over two years (mean±SD = 0.63±58.6 nmol $m^{-2}$ $h^{-1}$), and in October (mean±SD = 10.2±14.7 nmol $m^{-2}$ $h^{-1}$). We are thus confident that the low fluxes measured using the automatic chambers are real and that an insufficient closure time is not the reason for the low fluxes.

The low $R^2$ values can be due to different reasons, high variability during closure time or small fluxes (see point 4). We would like to point out that the static chamber fluxes show rather low $R^2$ values too. However, we found a clear positive relationship between the $R^2$ value and the magnitude of the flux for both measurement techniques. Therefore, we suggest looking at a different criterion to assess the $N_2O$ flux quality, such as the mean square error (RMSE) which is not dependent on the flux magnitude itself.

4) At the site, N supply to plants and microorganisms is limited, so it is to be expected that $N_2O$ fluxes are low as well. Foliage N concentrations indicate N limitation for spruce (foliar N concentration are about 1% in 0- and 1-yr-old needles as opposed to the optimum range of N content in needles between 1.5 and 2.3 %; Thimonier et al., 2010; Ingestad, 1959), N concentrations in the soil are low (1.4% in the organic layer and 0.4% in 10-20 cm depth, Jörg, 2008), N deposition is low (about 10 kg N $ha^{-1}$ $yr^{-1}$; Thimonier et al., 2019; Gharun et al., 2021). Thus, our site is rather low in N which could be used for microbial transformations, competing with plant uptake (Schulze et al. 2019).

Moreover, we measured $N_2O$ fluxes using eddy covariance above the forest in 2016 and 2017 where we found that the forest emitted 0.047 g $N_2O$ $m^{-2}$ $yr^{-1}$ which corresponds to 0.122 μmol $N_2O$ $m^{-2}$ $h^{-1}$. These low fluxes further back up the notion that the soil $N_2O$ fluxes are very low at the site. Furthermore, a study conducted a boreal spruce forest with low nitrogen deposition rates (about 5 kg N $ha^{-1}$ $yr^{-1}$), reported very low mean $N_2O$ fluxes of around 0.02 μmol $N_2O$ $m^{-2}$ $h^{-1}$ agreeing well with our results (Rütting et al., 2021).

5) The flux detection limit has been assessed based on the precision of the laser and the closure time. Our flux detection limit is a lower bound, i.e., under ideal conditions, we could detect fluxes as low as 29 nmol $N_2O$ $m^{-2}$ $h^{-1}$. The fluxes within the range of ±29 nmol $N_2O$ $m^{-2}$ $h^{-1}$ are therefore not significantly different from zero. It is correct though, that the real measurement precision is likely to be higher than 0.03 ppb. This basically just means that our flux detection limit would be higher. A study by Nemitz et al. (2018) reports the precision of the instrument as 0.09 ppb. This would give a flux detection limit of 87.2 nmol $N_2O$ $m^{-2}$ $h^{-1}$, which is still a very low flux. The flux detection limit could be reduced with longer chamber closure time. According to this logic, we can say that the higher the measured flux, the

more confidence we have in the measurements. Overall, we can say that we are confident that our $N_2O$ measurements are very low and not relevant for the GHG budget of the forest floor.

Based on the explanations given above, we would like to keep and discuss the $N_2O$ fluxes in the manuscript. Long-term and high-resolution $N_2O$ chamber measurements in forest ecosystems are very rare which is why we think showing this dataset is very important for the scientific discourse. We would like to redo the quality assessment of the $N_2O$ fluxes using the RMSE instead of the $R^2$ which is dependent on the slope of the linear regression. Generally, we suggest moving the $N_2O$ figure (Fig. A.1) from the appendix to the main text, add a panel (d) showing the boxplot from the static chamber measurements (Fig. 1), and discuss these measurements more in detail later on.

[Figure]

Fig. 1: Forest-floor $N_2O$ fluxes (nmol m$^{-2}$ h$^{-1}$) for the years 2017 (a) and 2020 (b). Black lines show means over four chambers, grey bands show standard deviations among four chambers. Boxplot showing distribution of means over four automatic chambers (c) and $N_2O$ fluxes from static chamber measurements (d). The dotted lines depict the minimum flux which could be detected by the Dual Quantum Cascade Laser spectrometer.

*Some information should be added to the method section about how the snow in the chamber was treated in flux calculations. The volume of the water(ice) of the snow cover must be subtracted from the camber volume during snow cover. How was that done? Was snow porosity measured or assumed somehow? If the snow volume was not subtracted, it is no wonder that CO2 emissions became the lower the more snow was in the chamber. Or was only the volume above the snow surface used for calculation? (In this case it would be flux from the snow surface, not the forest floor). Just of interest, what had happened in spring? Typically opaque chambers warm up faster and snow melts much earlier in and around them.*

Thanks for the comment. Indeed, we did not provide enough details on how we treated the presence of snow in the chamber during flux calculations. As the reviewer rightly guessed, we subtracted the snow volume from the chamber volume during snow covered periods. We also accounted for the varying chamber volume due to the chamber frames we had installed during winter. Thus, the original flux calculations were correct. We will add the following formula to the manuscript, which describes the volume calculation for the flux calculations:

V = 0.75 * 0.75 * (0.5 + frame number * 0.5 – snow depth)

Where the chamber area was 0.75 m * 0.75 m. The height of the chamber was 0.5 m. However, the height of the volume depended on the number of additional frames added to allow flux measurements during snow covered periods as well as the snow depth inside the chamber. The additional frames were made of the same white PVC as the chambers themselves with a height of 0.5 m. Up to two additional frames were added.

With our chamber design we are confident to have avoided potential side effects on the environmental conditions as much as possible. The chambers were white (high albedo), very large (reducing edge effects), and in the open position, they moved far away from the soil collar (avoiding shading). We do not have data on the timing of snow melt inside of the chambers. However, we observed that snow melted rather later inside the chamber compared to outside. To assess potential chamber effects on environmental conditions, we measured soil temperature inside and outside of each of the four chambers. We did not find any significant differences inside vs. outside the chambers (Fig. 2a). In winter (Dec-Mar), the temperatures inside the chambers (orange) were only 0.1-0.5 ºC lower than outside the chambers (blue, depending on the chamber). However, the differences were only significantly different from zero in December and February-March (Fig. 2b). In these months, soil temperatures are around 0 °C. When looking at the temperature response curve of forest-floor respiration (Fig. 2c), we can say that at such low soil temperatures, a difference of 0.5 °C in soil temperature has a minimal effect on the magnitude of forest-floor respiration. Therefore, we think that the effect of our chambers on the forest-floor respiration is negligible. We will add this info in the appendix of the revised manuscript.

[Figure]

Fig. 2: a) Soil temperatures ($T_{soil}$) at 5 cm inside (orange) and outside (lightblue) the chambers over the course of a year. b) Difference in $T_{soil}$ at 5 cm between inside and outside the chambers over the course of a year. Lines show means, bands show standard deviations over all chambers and three years (2017, 2020 and 2021). c) $Q_{10}$ model showing the relationship of daily means of $T_{soil}$ at 5 cm and forest-floor respiration of all chambers and years.

*It is often stated that high temporal resolution measurements have the advantage to capture hot-moments in GHG fluxes. Well – have such hot moments been observed? Currently it does not really seem so. Did freeze-thaw periods occur? What happened during these periods with CH4 fluxes? It would be important to zoom out some such hot moments from the long-term datasets and show them separately. Even if there was freeze-thaw and no peak in CH4 flux occurred – this could be shown. Were there any CH4 emissions at all, at least short-term? We for instance once observed a small CH4 and huge N2O peak during freeze-thaw in a deciduous forest (Schindlbacher Biogeochemistry 2022)*

*but very similar CH4 pattern with just lower uptake during winter as in your study (Heinzle AgrForMet 2023) in a spruce mountain forest. Another possibility for hot moments is after rain post drought periods. Did you observe any flux peaks of any GHG during such periods? Seems there were some very short term peaks and some longer-term positive CH4 fluxes in the "observed fluxes" in Fig. A.3.*

Thanks for the suggestion to look deeper into hot moments. We actually have already described bursts of $CH_4$ emissions coinciding with snow fall and snow melt in section 3.1 of the original manuscript. Furthermore, we showed daily flux data in Fig. 1 of the originally submitted manuscript where such emission periods can be observed. Since the manuscript is already quite dense, we had planned to write a separate paper concerning short-term variations in $CH_4$ as well as $CO_2$ fluxes, using the full resolution of the data (3-hourly) and showing the data per chamber separately. In the next manuscript, we would like to not only investigate the short-term effects of snowfall, snow melt and freeze-thaw events, but also effects of drought and precipitation (e.g., rewetting events) on the fluxes. Here, we focused on daily flux data as well as long-term means over all chambers to represent the entire forest. We will rephrase the sections in the manuscript where we talk about hot moments.

*There appeared very high CO2 fluxes during a period in summer 2022 – any idea why? If possible the advantages of the high resolution GHG dataset should be worked out – but probably the advantage is not as great as can be seen from the fact that the simple $Q_{10}$ driven model produced more or less the same CO2 budgets as the random forest model with a lot more input parameters than soil temperature.*

Thank you for the comment about the very high $CO_2$ fluxes in 2022. As shown in the manuscript, summer 2022 temperatures were very high at the site; summer 2022 was the warmest summer since we started our measurements at the site in 1997. It is well known that temperature is a major driver for any respiratory process (Davidson et al., 2006; Amthor, 2000). Also our Random Forest (RF) driver analysis revealed that soil temperature was the main driver for forest floor $CO_2$ fluxes. Furthermore, a study by Anjileli et al. (2021) has shown that heat extremes can increase the soil respiration by 25%. Therefore, it is not surprising that we measured very high forest floor $CO_2$ fluxes during summer 2022.

Since we found that the $CO_2$ fluxes at our site are mainly driven by soil temperature, it is not surprising that the $Q_{10}$ model worked reasonably well ($R^2 = 0.86$) and generally captured the temporal dynamics of respiration at our site. However, we would like to point out that this is not self-evident. Many studies have shown that $Q_{10}$ models do not reproduce measured fluxes well when additional drivers impact the fluxes, for instance when soil moisture, frost, or carbohydrate limitations come into play (e.g., Rühr and Eugster, 2009; Reichstein et al., 2013; Mitra et al., 2019).

In our study, the very high fluxes in summer 2022 were not accurately reproduced by the $Q_{10}$ model, while the RF model estimated them well. Therefore, we argue that $Q_{10}$ models are not able to capture extreme fluxes which might be caused by more drivers than temperature alone. In contrast, our high-resolution dataset coupled with machine learning offered a more comprehensive model, which included multiple environmental variables and at the same time was able to consider chamber specific characteristics, and thus was able to capture the extreme fluxes. Thus, we think that the reliability of the RF budget is higher compared to the $Q_{10}$ budget. In order to make this point clearer, we will also discuss this in more detail in the revised manuscript.

Moreover, unlike $CO_2$ fluxes, $CH_4$ fluxes cannot be effectively modelled using a simple and reliable approach like the $Q_{10}$ model. Therefore, we favor high-resolution data and machine learning approaches to gain insights into the underlying drivers and obtain reliable GHG budget estimates.

**2. Line Comments**

**1.1 Title**

*"Continous" is a bit confusing since there were no measurements done in 2018 and 2019*

We rephrased the title to: "Forest-floor respiration, $N_2O$ and $CH_4$ fluxes in a subalpine spruce forest: Drivers and annual budgets".

**1.2 Intro: P2 50-60:**

*I would rather not discuss soil warming experiments here. The current study is no soil warming experiment and has no connection. You might better refer to generally increasing soil temperatures (Lembrechts, J. J., (2022) Global Change Biology, 28(9), 3110-3144.) and to the global trend of soil respiration under warming (Jian, Jinshi, et al. "A restructured and updated global soil respiration database (SRDB-V5)." Earth System Science Data 13.2 (2021): 255-267.; Bond-Lamberty, Ben, and Allison Thomson. "Temperature-associated increases in the global soil respiration record." Nature 464.7288 (2010): 579-582.).*

We agree that soil warming experiments are not relevant to the current study. We will change this part of the introduction and incorporate the mentioned references in the introduction.

**1.3 Methods:**

*Calibration of laser?*

The instrument has not been calibrated during the measurement campaign. However, to ensure fit quality, the laser temperatures and tuning rates were adjusted on a regular basis. After the measurement campaign in 2017, the instrument was sent to Aerodyne laser instrument for repair and maintenance; the laser measuring $N_2O$ was replaced in 2019. We did not calibrate the instrument since we were not interested in absolute concentrations. Furthermore, comparable instruments used for $N_2O$ measurements with eddy covariance are commonly not calibrated on a regular basis either. In a study by Rannik et al. (2015) comparing the available equipment for $N_2O$ flux measurements employing the EC technique and evaluating their performance, ability to detect small fluxes, and assessing long-term stability in determining the $N_2O$ exchange with a comparable instrument (CW-TILDAS-CS, Aerodyne), no calibration was performed during the campaign.

**1.4 Methods: Tab1:**

*For snow depth usually rather the maximal depth is provided than the mean depth*

Thank you for the comment, we will add the maximum depth to the table.

**1.5  Methods: Line 140**

*It is mentioned that 2% of the data were discarded after step 1 but not how many data were discarded after step 2 and 3. Please add.*

Step 2 removed 0.2% and 0.7% of $CO_2$ and $CH_4$ fluxes, respectively. Step 3 excluded 6% and 9% of $CO_2$ and $CH_4$ fluxes, respectively. We will add this information to the manuscript.

**1.6  Fig.1**

*Fig 1 and similar cases: It took me a while until I figured out that the record is not consecutive and that 1.1.2020 follows the 31.12.2017. Please indicate somehow that there is a 2 year gap in the dataset (eg. by a gap between the panels)*

Thank you for the comment. We will add a gap between the panels as suggested and update the figure caption as well. Following the updated figure:

[Figure]

Fig. 3: Daily mean a) air temperature and soil temperature at 5 cm depth, b) water-filled pore space at 5 cm depth (left axis) and daily sum of precipitation (right axis), c) snow depth, and daily mean forest-floor d) respiration fluxes (not gap-filled), and e) $CH_4$ fluxes (not gap-filled), for the years 2017, 2020, 2021, and 2022. Please note the gap in measurements between 2017 and 2020. Black lines show means over four chambers, grey bands show standard deviations among four chambers. All data shown were quality-checked as described in the main text.

**1.7 Line 230**

*I'd rather write "after snowmelt" instead of late winter*

We are not sure what the reviewer refers to since we did not use the expression "late winter" in line 230. However, did you mean line 232 where we wrote "at the end of winter"? This we can change to "after snowmelt".

**1.8 Lines 300-305**

*When discussing the budgets terms such as "higher-than-usual" "considerably higher" etc. should be avoided. If you want to make a solid statement that the 2022 fluxes were higher than average, then it would be necessary to apply statistics to the budget data. Otherwise no scientifically valid conclusion can be drawn.*

Thank you for the comment. We will back up our statements with statistics in a revised manuscript. We will add to the text that the 2022 $CO_2$ budget and the annual mean $T_{soil}$ of 2022 fall outside of the 95% confidence intervals ($\pm 1.96SD$, i.e., for the $CO_2$ budget: $\pm 392$ g $CO_2$ m$^{-2}$ yr$^{-1}$). In case of $CH_4$, the 2022 budget does not fall outside of the 95% confidence interval, therefore, we will delete the word "considerably" in line 302.

**1.9 Line 305**

*delete "exceptionally"*

We will change this, thank you.

**1.10 Discussion: Line 315**

*There is a bulk of literature that shows that CO2 emissions are depressed under really dry conditions – please rephrase here (the WFPS in the manuscript are generally rather low, but this is likely a matter of the very low bulk densities, which might have been underestimated a bit?). Anyway, was the soil in summer 2022 really so dry?*

Thanks for the comment. We would like to point out that it is difficult to obtain reliable absolute values of soil moisture at our site due to a very heterogeneous soil, and many roots and rocks in the upper horizons (please see our response on SWC to reviewer 2). Therefore, in our discussion we would rather not focus on absolute SWC or WFPS values, but rather on relative changes in soil moisture. Annual means of WPFS were indeed lowest for 2022 compared to the other years, but also soil temperatures were very high (see above). We do not see any indication of soil moisture limitation of respiration in

our data (see driver analysis). Bulk densities are indeed low at the site which is likely due to high organic matter content in the upper 5 cm. Bulk density calculations were done using soil data collected according to ICOS RI standards which gives us high trust that these values are correct (Arrouays et al., 2018, please see also Saby et al., 2023). We will rewrite the mentioned text section.

**1.11 Table 4**

*Table 4 can be moved into the supplements, or also the results = measured fluxes from all the studies are shown in the table for comparison with the current ones.*

Since long-term measurements for multiple greenhouse gas emissions are scarce, we would like to add the flux rates to Table 4 and keep it in the main text.

**3. References**

Amthor, J. S.: The McCree–de Wit–Penning de Vries–Thornley Respiration Paradigms: 30 Years Later, Ann. Bot., 86, 1–20, https://doi.org/10.1006/anbo.2000.1175, 2000.

Anjileli, H., Huning, L. S., Moftakhari, H., Ashraf, S., Asanjan, A. A., Norouzi, H., and AghaKouchak, A.: Extreme heat events heighten soil respiration, Sci. Rep., 11, 6632, https://doi.org/10.1038/s41598-021-85764-8, 2021.

Arrouays, D., Saby, N. P. A., Boukir, H., Jolivet, C., Ratié, C., Schrumpf, M., Merbold, L., Gielen, B., Gogo, S., Delpierre, N., Vincent, G., Klumpp, K., and Loustau, D.: Soil sampling and preparation for monitoring soil carbon, Int. Agrophysics, 32, 633–643, https://doi.org/10.1515/intag-2017-0047, 2018.

Davidson, E. A. and Janssens, I. A.: Temperature sensitivity of soil carbon decomposition and feedbacks to climate change, Nature, 440, 165–173, https://doi.org/10.1038/nature04514, 2006.

Gharun, M., Klesse, S., Tomlinson, G., Waldner, P., Stocker, B., Rihm, B., Siegwolf, R., and Buchmann, N.: Effect of nitrogen deposition on centennial forest water-use efficiency, Environ. Res. Lett., 16, 114036, https://doi.org/10.1088/1748-9326/ac30f9, 2021.

Hutchinson, G. L. and Mosier, A. R.: Improved Soil Cover Method for Field Measurement of Nitrous Oxide Fluxes, SOIL SCI SOC AM J, 45, 1981.

Ingestad, T.: Studies on the Nutrition of Forest Tree Seedlings. II Mineral Nutrition of Spruce, Physiol. Plant., 12, 568–593, 1959.

Mitra, B., Miao, G., Minick, K., McNulty, S. G., Sun, G., Gavazzi, M., King, J. S., and Noormets, A.: Disentangling the Effects of Temperature, Moisture, and Substrate Availability on Soil $CO_2$ Efflux, J. Geophys. Res. Biogeosciences, 124, 2060–2075, https://doi.org/10.1029/2019JG005148, 2019.

Nemitz, E., Mammarella, I., Ibrom, A., Aurela, M., Burba, G. G., Dengel, S., Gielen, B., Grelle, A., Heinesch, B., Herbst, M., Hörtnagl, L., Klemedtsson, L., Lindroth, A., Lohila, A., McDermitt, D. K., Meier, P., Merbold, L., Nelson, D., Nicolini, G., Nilsson, M. B., Peltola, O., Rinne, J., and Zahniser, M.: Standardisation of eddy-covariance flux measurements of methane and nitrous oxide, Int.

Agrophysics, 32, 517–549, https://doi.org/10.1515/intag-2017-0042, 2018.

Rannik, Ü., Haapanala, S., Shurpali, N. J., Mammarella, I., Lind, S., Hyvönen, N., Peltola, O., Zahniser, M., Martikainen, P. J., and Vesala, T.: Intercomparison of fast response commercial gas analysers for nitrous oxide flux measurements under field conditions, Biogeosciences, 12, 415–432, https://doi.org/10.5194/bg-12-415-2015, 2015.

Reichstein, M., Bahn, M., Ciais, P., Frank, D., Mahecha, M. D., Seneviratne, S. I., Zscheischler, J., Beer, C., Buchmann, N., Frank, D. C., Papale, D., Rammig, A., Smith, P., Thonicke, K., van der Velde, M., Vicca, S., Walz, A., and Wattenbach, M.: Climate extremes and the carbon cycle, Nature, 500, 287–295, https://doi.org/10.1038/nature12350, 2013.

Rühr, N. K. and Eugster, W.: Soil respiration fluxes and carbon sequestration of two mountain forests in Switzerland, 2009.

Rütting, T., Björsne, A.-K., Weslien, P., Kasimir, Å., and Klemedtsson, L.: Low nitrous oxide emissions in a boreal spruce forest soil, despite long-term fertilization, Front. For. Glob. Change, 4, 710574, https://doi.org/10.3389/ffgc.2021.710574, 2021.

Saby, N., Loubet, B., Goydarag, M. G., Papale, D., Arrouays, D., and Lafont, S.: Computing C Stock for one ICOS Site, ICOS Ecosystem Thematic Centre, 2023.

Thimonier, A., Graf Pannatier, E., Schmitt, M., Waldner, P., Walthert, L., Schleppi, P., Dobbertin, M., and Kräuchi, N.: Does exceeding the critical loads for nitrogen alter nitrate leaching, the nutrient status of trees and their crown condition at Swiss Long-term Forest Ecosystem Research (LWF) sites?, Eur. J. For. Res., 129, 443–461, https://doi.org/10.1007/s10342-009-0328-9, 2010.

Thimonier, A., Kosonen, Z., Braun, S., Rihm, B., Schleppi, P., Schmitt, M., Seitler, E., Waldner, P., and Thöni, L.: Total deposition of nitrogen in Swiss forests: Comparison of assessment methods and evaluation of changes over two decades, Atmos. Environ., 198, 335–350, https://doi.org/10.1016/j.atmosenv.2018.10.051, 2019.

---

## Author Comment (AC2)

**Discussion of "Forest-floor greenhouse gas fluxes in a subalpine spruce forest: Continuous multi-year measurements, drivers, and budgets"**

**Author Response to Referee 2 comments**

Krebs et al.

November 30, 2023

In the following, *reviewer comments are given italics*, author comments are given in normal font.

**1. General comments**

*This ms reports high-resolution GHG fluxes from a forest floor in a subalpine coniferous forest using four automated chambers. Such automatic measuring systems are of great scientific interest, because events that occur for a short time can be recorded with them. The GHG measurements are integrated in a network for long-term observations of ecosystem fluxes.*

Thanks for the comment.

*Three objectives were defined, but no hypotheses or research questions.*

This is correct. In the revised manuscript, we will add the following hypotheses:

"We hypothesize that the forest-floor is a source of $CO_2$ throughout the years, with large seasonal variability due to the temperature sensitivity of respiratory processes, but with very low $N_2O$ emissions due to the overall low N supply at the site. In contrast, we expect that the forest-floor is a net sink of $CH_4$, with soil temperature and snow dynamics being important drivers due to their impact on microbial activity and diffusion rates between soil and atmosphere. Thus, we expect the highest respiratory $CO_2$ emissions and highest $CH_4$ uptake in exceptionally warm years, such as in 2022 at our site. Overall, we anticipate the GHG budget being mainly determined by $CO_2$ fluxes, with $CH_4$ uptake only slightly offsetting the respiratory $CO_2$ losses, and very low $N_2O$ emissions."

*The measurement technique of CO2 and CH4 fluxes seems very robust, while the measurement technique for N2O fluxes is obviously critical. Many N2O measurements indicate negative values, a net N2O uptake by the forest floor. Few net N2O uptakes have been reported, but mostly in dry soils during the summer months. I can only speculate that the measurement duration of 180 seconds is too short for the large chamber volume (281 L) or for the height of the chambers (50 cm) at low N2O fluxes. Own measurements in a spruce forest with a different laser technique and a different chamber system showed that the measurement time often required more than 20 min before a significant increase of the N2O concentration could be determined. In this respect, I propose to remove the N2O measurements completely from the manuscript and focus on CO2 and CH4 fluxes.*

The suggestion to remove the $N_2O$ fluxes from the manuscript seems to be based on two reasons:
1) measurement duration of 180 sec is too short for the large chamber volume, and
2) unlikely that forest floor shows uptake of $N_2O$.
In the following, we want to address those two points.
1) In order to check for the validity of our chamber measurements, we have performed measurements using static chambers with the dimensions of d = 30 cm and h = 30 cm (Hutchinson and Mosier, 1981). We used eight static chambers, i.e., four chambers next to the automatic chambers, and four chambers placed randomly within the research area. Soil collars were installed two weeks prior to the first measurement campaign. Four rounds of sampling were done on two measurement days in October 2023 (n=32), when soil temperatures were between 5.5-10 °C, well above the long-term mean, and soil moisture values above 8%, favoring microbial activities. Three collars were irrigated between the first and second sampling round on the two days to simulate a heavy rainfall event, favoring denitrification. We left the chambers closed for 1 h and sampled the air in the headspace every 20 min. The fluxes that we measured with the eight static chambers were low (mean±SD = 2.9±31.1 nmol $m^{-2}$ $h^{-1}$). Furthermore, they agreed very well with the fluxes which we have measured using the automatic chambers over two years (mean±SD = 0.63±58.6 nmol $m^{-2}$ $h^{-1}$), and in October (mean±SD = 10.2±14.7 nmol $m^{-2}$ $h^{-1}$). We are thus confident that the low fluxes measured using the automatic chambers are real and that an insufficient closure time is not the reason for the low fluxes.
2) The $N_2O$ fluxes measured with the static chambers mentioned above showed occasional $N_2O$ uptake as did the automatic chamber measurements. We would like to point out that the uptake rates we have measured are very low and probably not significantly different from zero. However, microbial processes in forest soils can contribute to both uptake and release of $N_2O$, depending on the prevailing environmental conditions such as oxygen availability, soil moisture and microbial communities. Under aerobic conditions, denitrification contributes to $N_2O$ release, while under aerobic conditions, $N_2O$ reduction to $N_2$ can dominate over $N_2O$ production, which results in observations of net $N_2O$ uptake by soils (Wen et al., 2017). Moreover, $N_2O$ uptake has been observed in a German spruce forest (Goldberg and Gebauer, 2009). Therefore, we think occasional $N_2O$ uptake as measured with our chambers are real.

Due to the scarcity of long-term and high-resolution $N_2O$ fluxes from forest ecosystems, we think that our dataset is very valuable and would therefore like to keep the $N_2O$ fluxes in the manuscript. Instead of showing the $N_2O$ fluxes only in the appendix, we would like to move them to the main text (including the data from the static chamber measurements) and discuss them in the discussion part. Please see our response to your comment on the discussion part (section 1.12 Discussion) and our response to the comments on $N_2O$ fluxes of Referee 1.

*Another problem with respect to the calculation of the GHG budget is the contribution of ground vegetation to CO2 fluxes. Due to the opaque chambers, only the respiration of the vegetation is measured, as it naturally occurs only at night. Thus, CO2 fluxes were overestimated during daylight hours. For a correct GHG budget, however, the CO2 fixation of plants would also have to be recorded. An estimation of the contribution of aboveground plant organs to the CO2 flux would be interesting. Calculating the GHG budget for the forest does not seem justified to me.*

We agree that our budgets do not include $CO_2$ uptake from the understory plants during daytime and thus talking about a full forest-floor $CO_2$ budget is misleading. Thus, we adjusted our terminology and now talk about a forest-floor "respiration" budget when talking about $CO_2$ throughout the manuscript.

*Overall, a thorough revision of the manuscript is needed. As is usual in scientific papers, clearly formulated research questions or hypotheses, e.g. on the effect of the snowpack, would improve the quality of the ms.*

Thanks for your suggestions to improve the manuscript. On the hypotheses, see above. We hope we have addressed your overall concerns.

**2. Specific comments**

**1.1 Title**

*Please change the title if N2O fluxes are omitted. 'multi-year' is a bit exaggerated when the fluxes were only measured for 3-4 years.*

We rephrased the title to: "Forest-floor respiration, $N_2O$ and $CH_4$ fluxes in a subalpine spruce forest: Drivers and annual budgets".

**1.2 Line 16**

*Please present only means of the annual fluxes.*

We will change this, thank you.

**1.3 Line 19**

*Provide here the mean CH4 flux, not the CO2 equivalent*

We will change this, thank you.

**1.4 Line 19-20**

*'driven mainly by snow depth' – do you mean that increasing snow depth reduced CH4 uptake? Is the relation between CH4 flux and snow depth significant?*

Our random forest (RF) driver analysis showed that snow depth had the highest importance in the RF model to predict $CH_4$ fluxes. We will add a plot showing the curvilinear relationship between $CH_4$ fluxes and snow depth in the revised manuscript. Please see the suggested figure (Fig. 3) and our response to comment about line 271 (page 7 of this response; in short: 2x "yes").

**1.5 Line 27-28**

*'with negative effects on its carbon sink behavior' the data don't show this, please omit the statement.*

We would like to keep this statement in the manuscript due to the following reasons: i) We have shown that the forest-floor respiration budget was highest in the warm year of 2022. In the future, the forest site is projected to experience more years similar to 2022 (IPCC, 2021; CH2018, 2018). Thus, high respiratory losses from the forest floor will decrease the forest C sink. ii) Furthermore, studies show that the length of snow-covered periods will decrease in the Swiss Alps. This will also increase respiration fluxes and also contribute to decreasing C sinks of the forest (Klein et al., 2016; CH2018, 2018).

**1.6 Line 54-58**

*Experimental soil warming was not investigated in this study, but annual variation of gas fluxes. A more general view at temperature influence would better fit this study.*

We agree that soil warming experiments are not relevant to the current study. We will adjust this part of the introduction.

**1.7 Line 92 (Table 1)**

*Provide some data of the forest floor and mineral soil: horizons, thickness, texture, stocks. Does bulk density (5 cm) refer to the mineral soil or forest floor? (see comment below)*

The bulk density at 5 cm refers to the upper 5 cm of the mineral soil, which is high in organic matter. The stocks have already been reported in Table 1 of the original manuscript. The horizons, thicknesses and textures were measured for two soil profiles within the study area, a chromic cambisol and a rustic podzol. We will report information on horizons, thicknesses, and textures from these two profiles in the revised manuscript. Furthermore, in the meantime additional soil data from the ICOS ETC became available, which will also be shown.

**1.8 Line 113**

*180 s measuring time - why where chambers closed for 10 min? When where concentrations measured during the 10 min? Please provide the length of the tubing between chamber and detector and the flow rate or pump rate.*

Measurement cycle: This is a misunderstanding. The complete chamber measurement cycle is 10 min, and this includes the time for closing and opening the chamber (the chamber moves very slowly, so it takes around 3.5 min to close and around 3.5 min to open the chamber). The time in which the chambers were actually closed was 3 min. During the entire chamber cycle, the concentrations were measured continuously once per second. We will rephrase the text so that it becomes clearer.

The flow rate ranged between 0.9-1.0 slpm. The tube lengths between chamber and instruments ranged between 49-85 m. We determined the time lags in the arrival of the gas in the instrument based on the change in chamber status (fully open, fully closed) and max. $CO_2$ concentrations measured.

**1.9 Line119**

*Were the chambers closed 16 times per day = 160 min or 11% of daytime? Does this mean that 11% of annual precipitation was also excluded and the forest floor was drier than outside the chambers?*

We are aware that by using any chamber method, we are potentially altering environmental conditions. This is unavoidable for all chamber studies. However, with our chamber design and closure duration, such potential effects could be avoided as much as possible, since the chambers were white (high albedo), very large (reducing edge effects), and in the open position, they moved far away from the soil collar (avoiding shading; Fig. 1). See also our response on soil temperatures to referee 1.

We will include a picture of one of the chambers in the appendix (Fig. 1) to show how the chamber moves and that about 7 minutes of the 10 minute cycle were used to move the chamber down onto the frame. Thus, the chambers were actually only fully closed for 3 minutes per chamber cycle = 48 minutes or 3.3% of the day, and not 160 minutes per day. If we add the time spent opening and closing the chamber as it hovers over the frame (4 minutes per cycle), we estimate – very conservatively – that the chamber is closed for a maximum of 7 minutes per chamber cycle = 7.8% of the day. However, rain does not usually fall perpendicular to the floor, but at an angle, i.e., during these 4 minutes, rain will still fall inside the frame. We think that our conservative estimate of 7 minutes is thus more realistic than the 10 minutes assumed by referee 2. We will add this info into the Materials and Methods section in the revised manuscript.

[Figure]

Fig. 1: Picture of chamber 3.

Moreover, we think it is too simplistic to say that we exclude 11% or (see above) 7.8% of the precipitation, because the chambers were closed for 11 or 7.8% of the day. Rainfall is not evenly distributed throughout the day. Moreover, in a spruce forest, throughfall is typically less than bulk

precipitation above the canopy due to interception and is very heterogeneous within a forest (Schulze et al., 2019). These factors challenge the statement that we exclude a certain percentage of bulk precipitation because we close the chambers for this percentage of the day.

Furthermore, we have the chance to test for soil moisture bias due to the chambers because for our chambers 1 and 2, we do have soil water content (SWC) measurements from inside and outside the chambers available for four years (Fig. 2). SWC was highly variable over time as well as in space. SWC differences between inside and outside varied between plus 10% and minus 10% during the four years. No clear trend was detectable over time. The average difference between inside and outside SWC over the four years was -2.9±5.8%. During most of the year, no significant difference in SWC inside vs. outside the chamber was detected, although we found on average 5% lower SWC values inside the chamber during winter (Fig. 2b). Based on the rather large uncertainties in absolute measurements of SWC (see answer to comment on Line 155), we believe that a difference of 5% is minor. Moreover, we found a high agreement in the dynamics of SWC inside and outside the chambers 1 and 2 when applying a Pearson correlation of the SWC inside and outside the chambers ($R^2$ values of 0.69 and 0.82). We will add this information to the revised manuscript.

[Figure]

Fig. 2: a) Soil water content (SWC) at 5 cm inside (orange) and outside (lightblue) the chambers over the course of a year. b) Difference in SWC at 5 cm between inside and outside the chambers over the course of a year. Lines show means, bands show standard deviations over all years and chambers 1 and 2.

**1.10   Line 155**

*The installation depth was 5 cm for the SWC sensors. The low bulk density indicates that the sensors were installed in the organic horizon or in the transition from the organic horizon to the mineral A horizon. This is critical because the EC-5 sensors have only a standard calibration, which is often not suitable for many forest soil horizons with high root density or stone fraction. Where the sensors calibrated with the soil from 5 cm depth? While the sensors show nicely the dynamics of the water content, the absolute value is often incorrect. When bulk density changes due to shrinkage and swelling of the forest floor, further uncertainty is added to WFPS. Overall, the WFPS is very low (Fig. 1b), especially after snowmelt where much higher values should be reached.*

We fully agree that reliable absolute measurements of SWC are difficult to obtain. Especially at the Davos site where the soil is very heterogeneous, and the upper horizons are full of roots and rocks which makes reliable calibration impossible. Since we were aware of these aspects, we used centered and scaled WFPS values for our data analyses as described in the original manuscript. With this approach, we take the correct temporal dynamics into account but avoid relying on potentially incorrect absolute values.

**1.11 Line 271**

*This result could be better presented, perhaps by linear/non-linear relationship (decrease in CH4 uptake/cm snow depth)*

Thanks for this suggestion. We will add a plot showing the relationship between $CH_4$ uptake and snow depth in the revised manuscript (Fig. 3). Furthermore, we will add that the snow depth and the GHG fluxes are highly correlated in the months Oct-May, as spearman correlations coefficients are 0.59 and -0.79 for $CH_4$ fluxes and forest-floor respiration, respectively.

[Figure]

Fig. 3: Relationship between forest-floor $CH_4$ fluxes (nmol m$^{-2}$ s$^{-1}$) and snow depth (cm). Black line shows fitted logarithmic curve.

**1.12 Discussion**

*N2O fluxes are not discussed at all.*

It is true that we did not discuss the $N_2O$ fluxes in the manuscript. We decided to not use them for driver analysis and budget calculations, two main objectives of the manuscript. This decision was made mainly because of the low magnitude of the fluxes and their irrelevance for the forest-floor GHG budget (using the mean $N_2O$ flux measured with the automatic chambers over the two years, 0.63 nmol m$^{-2}$ h$^{-1}$, we arrive at an annual budget of 0.066 g $CO_2$-eq m$^{-2}$ yr$^{-1}$ which represents 0.003% of the annual forest-floor GHG budget). However, we still think that it is important to show the $N_2O$ fluxes in the manuscript

because such measurements in forests are very scarce. So, instead of removing $N_2O$ from the manuscript completely, we would like to move the $N_2O$ figure (Fig. A.1 in the submitted manuscript) to the main text and adding a panel (d) showing the fluxes from the static chamber measurements (Fig. 4). This allows us to discuss the $N_2O$ fluxes in the paper and show that the magnitude of the fluxes is indeed very low.

[Figure]

Fig. 4: Forest-floor $N_2O$ fluxes (nmol $m^{-2}$ $h^{-1}$) for the years 2017 (a) and 2020 (b). Black lines show means over four chambers, grey bands show standard deviations among four chambers. Boxplot showing distribution of means over four automatic chambers (c) and $N_2O$ fluxes from static chamber measurements (d). The dotted lines depict the minimum flux which could be detected by the Dual Quantum Cascade Laser spectrometer.

**1.13   Line 370**

*How many 'hot moments' were identified in this study. One message of this study could be that the effort with automatic measurement systems for these forest types is very large and weekly or bi-weekly measurements with many chambers yield more robust flux rates on a larger spatial scale.*

The question about "hot moments" is difficult to answer since we focused in our manuscript mainly on daily and annual fluxes, not necessarily on hot moments, even though we described and discussed them in the original manuscript.

Nevertheless, we do not think that automatic measurements always need more effort than manual bi/weekly measurements, which need more person-power than automatic chambers, particularly when visiting remote sites bi/weekly. The approach clearly depends on the research questions asked. However, hot moments can only be identified when high-temporal resolution measurements are available, which are very difficult to obtain in high enough temporal resolution with manual measurements. Those are typically taken during daytime and good weather conditions, rarely 24/7/365 as automatic measurements. We agree that manual measurements can represent spatial variability better than automatic measurements, which need mains power if run at high temporal resolution. But then, "hot spots", not hot moments would be the research question asked.

**1.14   Table 1**

*Temperature, WFPS and snow cover are presented in Fig. 1. If needed, annual means can be described in the text.*

Thank you for the comment. However, Fig. 1 shows aggregated data for the entire research area while Tab. 1 gives data separated for the different chambers which we treated as replicates and used for the driver analysis. Thus, we suggest moving Table 1 to the Appendix instead of deleting it.

**1.15   Table 4**

*Were the fluxes in these studies measured exclusively from forest floors where vegetation had not been removed? If present, the above-ground soil vegetation is very often removed by clipping to measure soil respiration. There are many more long-term studies where GHG fluxes were published in different papers from the same forest site. A table without CO2 and CH4 flow rates is redundant anyway.*

Thank you for the comment. We agree that there are many more studies measuring one of the three GHG, but we only selected those in which all three greenhouse gases were measured at the same time. We reported studies irrespective of whether vegetation was removed or not. We will include the magnitude of $CO_2$, $CH_4$ and $N_2O$ fluxes (including the fluxes from our study) as well as information about vegetation removal in the table in the revised manuscript. We will also highlight in the text that soil respiration and forest-floor respiration are not equal and cite relevant references, such as Barba et al. (2018). However, we would like to stick to our approach and focus on studies which show all three GHG fluxes measured at the same time (and thus been published in the same paper), to be able to compare our study and approach with their measurement method and frequency.

**3.  References**

Barba, J., Cueva, A., Bahn, M., Barron-Gafford, G. A., Bond-Lamberty, B., Hanson, P. J., Jaimes, A., Kulmala, L., Pumpanen, J., Scott, R. L., Wohlfahrt, G., and Vargas, R.: Comparing ecosystem and soil respiration: Review and key challenges of tower-based and soil measurements, Agric. For. Meteorol., 249, 434–443, https://doi.org/10.1016/j.agrformet.2017.10.028, 2018.

CH2018: CH2018 – Climate Scenarios for Switzerland, Technical Report, National Centre for Climate Services, Zurich, 2018.

Goldberg, S.D. and Gebauer, G.: $N_2O$ and NO fluxes between a Norway spruce forest soil and atmosphere as affected by prolonged summer drought, Soil Biology & Biochemistry 41, 1986–1995, 2009.

Hutchinson, G. L. and Mosier, A. R.: Improved Soil Cover Method for Field Measurement of Nitrous Oxide Fluxes, SOIL SCI SOC AM J, 45, 1981.

IPCC: Climate change 2021: The physical science basis. Contribution of working group I to the sixth assessment report of the Intergovernmental Panel on Climate Change, Cambridge University Press, Chambride, United Kingdom and New York, NY, USA, 2021.

Klein, G., Vitasse, Y., Rixen, C., Marty, C., and Rebetez, M.: Shorter snow cover duration since 1970 in the Swiss Alps due to earlier snowmelt more than to later snow onset, Clim. Change, 139, 637–649, https://doi.org/10.1007/s10584-016-1806-y, 2016.

Schulze, E.-D., Beck, E., Buchmann, N., Clemens, S., Müller-Hohenstein, K., and Scherer-Lorenzen, M.: Biogeochemical Fluxes in Terrestrial Ecosystems, in: Plant Ecology, edited by: Schulze, E.-D., Beck, E., Buchmann, N., Clemens, S., Müller-Hohenstein, K., and Scherer-Lorenzen, M., Springer, Berlin, Heidelberg, 529–577, https://doi.org/10.1007/978-3-662-56233-8_16, 2019.

Wen, Y., Corre, M. D., Schrell, W., and Veldkamp, E.: Gross $N_2O$ emission and gross $N_2O$ uptake in soils under temperate spruce and beech forests, Soil Biol. Biochem., 112, 228–236, https://doi.org/10.1016/j.soilbio.2017.05.011, 2017.

---

## Author Response (AR2)

**Discussion of "Forest-floor respiration, $N_2O$, and $CH_4$ fluxes in a subalpine spruce forest: Drivers and annual budgets"**

Krebs et al.

February 29, 2024

In the following, *reviewer comments are given italics*, author comments are given in normal font.

**Author Response to Referee 1 comments**

*Thank you for thoroughly revising the paper. It now reads superwell and is very interesting and informative. Especially the fact that soil CO2 efflux was very high in the warm year, albeit drying out soils is interesting. I also very much appreciate the environmental driver analysis via the random forest model. If authors are in the mood, I'd suggest a quick last round of fine-tuning.*

Thanks a lot for the positive feedback.

*I suggest being careful with the terms carbon-balance or carbon-budget and GHG-balance or GHG-budget throughout the text. For a carbon–balance only the carbon (C) in the CO2 and CH4 is relevant. Hence, only the annual carbon and not the annual CO2-, or the CO2-equivalent sum is relevant and should be reported (hence only CO2-C not all the mass of CO2 including two oxygen molecules is the topic of the carbon balance). This problem for instance becomes evident in the Abstract (L20-25). What is reported here is to my understanding the GHG-budget. In the GHG-budget, the fluxes of CH4 (and N2O) are aligned with the CO2 fluxes by calculating their CO2-equivalents or global warming potentials. This, however, has nothing to do with a "carbon balance". Please check through the text, especially Page 17 4.3 Forest-floor C and GHG budgets.*

Thank you for the comment. We were not using the term "balance" in the earlier manuscript, so this might have been a misunderstanding/interpretation.

When it comes to the terms "carbon budget" and "GHG budget" and their respective units (C or $CO_2$-eq) when referring to $CO_2$ and $CH_4$ fluxes, it is true that we have not used them consistently enough throughout the whole text. We changed the text accordingly to make it more consistent. We now only write about a "GHG budget" when considering all three GHGs (in g $CO_2$-eq m$^{-2}$ s$^{-1}$). When not writing about all three, we mention the gases separately.

*The abstract reads superwell except for the last lines. The sentence mentioned above might be re-formulated. E.g "The mean forest floor GHG-budget indicated emissions of ....CO2-eq...., with respiratory fluxes dominating and CH4 uptake offsetting a small portion (0.8%) of the CO2 emissions." In the last sentence you may change to "....effects on the carbon sink of the forest ecosystem" (as it is written in the conclusions).*

Thank you for the comment. We changed the two sentences in the abstract accordingly. They now read:

"The mean forest-floor GHG budget indicated emissions of 2319±200 g $CO_2$-eq m$^{-2}$ yr$^{-1}$ (mean ± standard deviation over all years), with respiration fluxes dominating and $CH_4$ offsetting a very small proportion (0.8%) of the $CO_2$ emissions. …..... In a future with increasing temperatures and less snow

cover due to climate change, we expect increased forest-floor respiration at this subalpine site modulating the carbon sink of the forest ecosystem."

Similarly, we adjusted the text in 3.3 and Tab. 1 in the Results, and in the Discussion sections.

*It is true that there exist only hand full studies about year round GHG flux measurements in alpine mountain forests with winter snow-cover. Therefore the study of Heinzle et al. 2023 (Soil CH4 and N2O response diminishes during decadal soil warming in a temperate mountain forest in AgrForMet) might be considered for the discussion. Heinzle et al also observed low CH4 emissions during snow cover (fig4), which is in line with your results and the N2O fluxes might be used as a good example for higher fluxes at such higher N containing sites/soils. I don't want to push this particular study into your paper, but as mentioned above, such studies are rare and the few ones conducted might not be neglected. However, if you don't like the study out of any reason, I am totally fine if you do not cite it!*

Thank you for bringing this study to our attention. We included the paper in the sections 4.2 and 4.3.

**Author Response to Referee 2 comments**

*The authors have satisfactorily revised most parts of the manuscript. Nevertheless, there are few statements with which I do not agree or of which I am not convinced.*

*The hypothesis is a bit trivial. One reason for using the automatic chamber system was apparently the temporal variability of GHG fluxes including 'hot moments' which can have a large impact on the annual GHG budget. Even if no hot GHG moments were observed, the method is suitable for this purpose.*

We added the hypotheses on demand during the review process. We think that coming up with hypotheses in hindsight is not ideal, we would have preferred to stay with objectives only. "Hot moments" are not mentioned in the current version of the manuscript. We agree with the reviewer that this is a "hot topic", but this will be studied/written in a separate manuscript, although the reviewer seemingly would have loved to read about it in this manuscript already.

*I cannot agree with the statement that the N input by deposition and the N availability are low in the forest. N deposition rates have decreased in European forests over the past two decades but are still at a high level. The natural background of N deposition would be about 1-2 kg ha-1 y-1. With a deposition of 10 kg N ha-1 y-1, I would not expect any substantial N limitation in the coniferous forest. N2O emissions from coniferous forest soils are usually very low, even at significantly higher N inputs. This 'forest type' effect on N2O could be discussed in more detail.*

We politely disagree. The vegetation itself is the indicator for high or low N supply and N limitation. As mentioned in the manuscript, the foliar N concentration at the site is about 1 % which is well below the optimum range in needles of between 1.5 and 2.3 %. Moreover, N deposition per se is insufficient to detect high or low N supply, as nicely summarized by Butterbach-Bahl et al. (2011) in the European Nitrogen Assessment (Tab. 6.2, Chapter 6; Sutton et al., 2011):

**Table 6.2** Characteristics of coniferous forest ecosystems with low, intermediate and high N status (Gundersen *et al.*, 2006). Nitrogen input is not a good indicator of N status, but the ranges given are typical for low, intermediate and high N status ecosystems

| Nitrogen status | Low N status (N-limited) | Intermediate | High N status (N-saturated) |
|---|---|---|---|
| Input (kg N ha$^{-1}$ yr$^{-1}$) | 0–15 | 15–40 | 40–100 |
| Needle N% (in spruce) | < 1.4 | 1.4–1.7 | 1.7–2.5 |
| C:N ratio (g C g N$^{-1}$) | > 30 | 25–30 | < 25 |
| Soil N flux density proxy (litterfall + throughfall) (kg N ha$^{-1}$ yr$^{-1}$) | < 60 | 60–80 | >80 |
| Proportion of input leached (%) | <10 | 0–60 | 30–100 |

Furthermore, N deposition rates of 1-2 kg N ha$^{-1}$ yr$^{-1}$ are rather preindustrial. And yes, also at our site, N deposition has decreased over the last decades (Gharun et al., 2021; from 17.5 kg N ha$^{-1}$ yr$^{-1}$ in the late 1980ties). Critical loads (as the reviewer seems to have in mind) of 1-2 kg N ha$^{-1}$ yr$^{-1}$ are now-a-days used for soft-water alpine lakes, tundra etc., while for fir and spruce forests, the critical load is about 10 to 15 kg N ha$^{-1}$ yr$^{-1}$. This is supported by a recent report by Hettelingh et al. (2017) on European Critical Loads (chapter about Switzerland, p. 177-190, written by Swiss Federal Office for the Environment) where N deposition levels below 10 kg N ha$^{-1}$ yr$^{-1}$ were set as the lower limit above which critical loads for nutrient N (CLnutN) were calculated. This evaluation is supported by Braun et al. (2017) for Switzerland, and Wang et al. (2022) for Europe. Only N deposition rates above 20–22 kg N ha$^{-1}$ yr$^{-1}$ were negatively related with basal area increments for Norway spruce, thus harmful, while below this N deposition growth increased, clearly showing N limitation to tree growth (Braun et al., 2017). Similarly, N deposition rates around 22 kg N ha$^{-1}$ yr$^{-1}$ had the highest positive effect on NEP (C sink) of forests across Europe (Wang et al., 2022), again, indicating N limitation below this level. Together with the best indicator, i.e., the vegetation and its foliar N concentration, we think that we can safely say that our forest is indeed low in N, also compared to other Swiss and European forests, and thus low N$_2$O emissions are to be expected. We have added information and the additional references in the discussion, which now reads:

"At our site, N supply to plants and microorganisms is limited. Foliage N concentrations indicate N limitation for spruce (foliar N concentration are about 1 % in 0- and 1-yr-old needles as opposed to the optimum range of N content in needles between 1.5 and 2.3 %; Thimonier et al., 2010; Ingestad, 1959). Furthermore, N concentrations in the soil are low (1.4% in the organic layer, 0.4% in 10–20 cm depth; Jörg, 2008). N deposition at the site (about 10 kg N ha$^{-1}$ yr$^{-1}$; Thimonier et al., 2019; Gharun et al., 2021) corresponds to the lower level of critical N loads for forests in Switzerland (Hettelingh et al., 2017), well below the N deposition negatively related to basal area increments for spruce (20–22 kg N ha$^{-1}$ year$^{-1}$; Braun et al., 2017) or that with the highest positive effect on net ecosystem productivity, i.e., the C sink, of forests across Europe (22 kg N ha$^{-1}$ yr$^{-1}$; Wang et al., 2022). Thus, our site can clearly be considered rather low in N, which could be used for microbial transformations like nitrification, competing with plant uptake (Schulze, 2000), therefore, low soil N$_2$O fluxes were to be expected."

*I agree that overall N2O fluxes are very low but suspect that many negative fluxes could be methodological. However, the finding that the forest soil can be a net N2O sink even at higher soil water contents is critical in my view. How can the strongly fluctuating and negative N2O fluxes in winter 2020 be explained? As far as I know, net N2O uptake in soils was only observed under dry conditions in summer.*

We never claimed that we found a net N$_2$O sink even at higher soil water contents. Fig. 2 clearly shows large variations in N$_2$O fluxes but generally a source of N$_2$O during winter. Overall, the N$_2$O source of the forest floor is very low, close to zero. In any case, we are aware of at least one study from a temperate forest which has found net N$_2$O uptake during winter (Heinzle et al., 2023). Furthermore, we have applied strict quality assessment based on the RMSE of the linear fit in change in N$_2$O concentration, which excluded around 25 % of all N$_2$O fluxes. We are confident that the reported fluxes are correct.

*The argument with the chamber comparison did not completely convince me. An open question is whether low negative N2O fluxes would occur with a longer detection time of e.g. 20-60 min as compared to 180 s. The accuracy of the measurement should increase significantly with the duration of the measuring time. Such a test could be easily performed and would increase the credibility of the method. For future N2O measurements with laser technology, it is a fundamental question how long detection time should be to generate robust results. Given the very large headspace volumes and the very long tubes between the chambers and the laser, it would be useful to systematically investigate the influence of the detection time on the N2O flux rate.*

We agree that such a test would be helpful when no additional, clear evidence is available that the N status of the forest is low and therefore also the $N_2O$ fluxes. Moreover, for a methodological paper, one would need to compare fluxes using different/multiple laser spectrometers, but such a study should be done at a site where larger $N_2O$ fluxes occur, such as grasslands or croplands, not at a low N supplied spruce forest. As we have written in the manuscript (and the earlier answers to the reviewers), the choice of closure time is a compromise between avoiding confounding effects on environmental conditions during chamber closure and the flux detection limit of our method. As we have described in our responses in the last round, a longer closure time would enable us to reduce the flux detection limit. However, for our purpose, we did not need this level of precision, as highlighted by the fact that the second chamber method (i.e., static chambers and gas chromatography, closure time of 1 h) confirmed the low magnitude of $N_2O$ fluxes at our site. The small chambers had a much longer integration time and nevertheless showed minimal fluxes even when environmental conditions were favourable for $N_2O$ production. Therefore, based on this additional evidence, we are confident that the low $N_2O$ fluxes that we measured with our automatic chambers are reliable.

*Response 1.5 'Thus, high respiratory losses from the forest floor will decrease the forest C sink.' Higher soil CO2 fluxes can also be caused by an increase in root respiration or in litter production. Based on soil respiration, no conclusions can be drawn about the C sink strength of forests. To answer this important question, long-term and comprehensive analyses of all forest C fluxes or C stocks in the biomass and in the soil are required. Please, omit statements 'forest C sinks'.*

We are not sure we understand the reviewer correctly. The forest C sink is the difference between GPP and $R_{eco}$. Beyond our study, the flux community has clear evidence that respiration, one part of the equation ($R_{eco}$), might indeed increase in the future due to higher temperatures. Forest-floor respiration (including root respiration) is a major component of $R_{eco}$. Litter production is not increasing respiration, unless litter is decomposed, which would then be included in forest floor respiration and in $R_{eco}$. Therefore, we think it is indeed feasible to say that the forest C sink might be modulated by an increasing forest-floor respiration (see also our answer to reviewer 1, abstract).

**References**

Braun, S., Schindler, C., and Rihm, B.: Growth trends of beech and Norway spruce in Switzerland: The role of nitrogen deposition, ozone, mineral nutrition and climate, Sci. Total Environ., 599–600, 637–646, https://doi.org/10.1016/j.scitotenv.2017.04.230, 2017.

Butterbach-Bahl, K., Gundersen, P., Ambus, P., Augustin, J., Beier, C., Boeckx, P., Dannenmann, M., Gimeno, B. S., Ibrom, A., Kiese, R., Kitzler, B., Rees, R. M., Smith, K. A., Stevens, C., Vesala, T., and Zechmeister-Boltenstern, S.: Nitrogen processes in terrestrial ecosystems, in: The European Nitrogen Assessment: Sources, Effects and Policy Perspectives, edited by: Bleeker, A., Grizzetti, B., Howard, C. M., Billen, G., van Grinsven, H., Erisman, J. W., Sutton, M. A., and Grennfelt, P., Cambridge University Press, Cambridge, 99–125, https://doi.org/10.1017/CBO9780511976988.009, 2011.

Gharun, M., Klesse, S., Tomlinson, G., Waldner, P., Stocker, B., Rihm, B., Siegwolf, R., and Buchmann, N.: Effect of nitrogen deposition on centennial forest water-use efficiency, Environ. Res. Lett., 16, 114036, https://doi.org/10.1088/1748-9326/ac30f9, 2021.

Heinzle, J., Kitzler, B., Zechmeister-Boltenstern, S., Tian, Y., Kwatcho Kengdo, S., Wanek, W., Borken, W., and Schindlbacher, A.: Soil CH4 and N2O response diminishes during decadal soil warming in a temperate mountain forest, Agric. For. Meteorol., 329, 109287, https://doi.org/10.1016/j.agrformet.2022.109287, 2023.

Hettelingh, J. P., Posch, M., and Slootweg, J.: European critical loads: database, biodiversity and ecosystems at risk : CCE Final Report 2017, Rijksinstituut voor Volksgezondheid en Milieu, Bilthoven, Netherlands, 2017.

Ingestad, T.: Studies on the Nutrition of Forest Tree Seedlings. II Mineral Nutrition of Spruce, Physiol. Plant., 568–593, 1959.

Schulze, E.-D. (Ed.)Caldwell, M. M., Heldmaier, G., Lange, O. L., Mooney, H. A., Schulze, E.-D., and Sommer, U.: Carbon and Nitrogen Cycling in European Forest Ecosystems, Springer Berlin Heidelberg, Berlin, Heidelberg, https://doi.org/10.1007/978-3-642-57219-7, 2000.

Sutton, M. A., Howard, C. M., Erisman, J. W., Billen, G., Bleeker, A., Grennfelt, P., van Grinsven, H., and Grizzetti, B. (Eds.): The European Nitrogen Assessment: Sources, Effects and Policy Perspectives, Cambridge University Press, Cambridge, https://doi.org/10.1017/CBO9780511976988, 2011.

Thimonier, A., Graf Pannatier, E., Schmitt, M., Waldner, P., Walthert, L., Schleppi, P., Dobbertin, M., and Kräuchi, N.: Does exceeding the critical loads for nitrogen alter nitrate leaching, the nutrient status of trees and their crown condition at Swiss Long-term Forest Ecosystem Research (LWF) sites?, Eur. J. For. Res., 443–461, https://doi.org/10.1007/s10342-009-0328-9, 2010.

Thimonier, A., Kosonen, Z., Braun, S., Rihm, B., Schleppi, P., Schmitt, M., Seitler, E., Waldner, P., and Thöni, L.: Total deposition of nitrogen in Swiss forests: comparison of assessment methods and evaluation of changes over two decades, Atmos. Environ., 335–350, https://doi.org/10.1016/j.atmosenv.2018.10.051, 2019.

Wang, Y.-R., Buchmann, N., Hessen, D. O., Stordal, F., Erisman, J. W., Vollsnes, A. V., Andersen, T., and Dolman, H.: Disentangling effects of natural and anthropogenic drivers on forest net ecosystem production, Sci. Total Environ., 839, 156326, https://doi.org/10.1016/j.scitotenv.2022.156326, 2022.